# Self-Distillation Amplifies Regularization in Hilbert Space

Hossein Mobahi♣    Mehrdad Farajtabar§    Peter L. Bartlett♣‡

hmobahi@google.com   farajtabar@google.com   bartlett@eecs.berkeley.edu

♣ Google Research, Mountain View, CA, USA
§ DeepMind, Mountain View, CA, USA
‡ EECS Dept., University of California at Berkeley, Berkeley, CA, USA

## Abstract

Knowledge distillation introduced in the deep learning context is a method to transfer knowledge from one architecture to another. In particular, when the architectures are identical, this is called self-distillation. The idea is to feed in predictions of the trained model as new target values for retraining (and iterate this loop possibly a few times). It has been empirically observed that the self-distilled model often achieves higher accuracy on held out data. Why this happens, however, has been a mystery: the self-distillation dynamics does not receive any new information about the task and solely evolves by looping over training. To the best of our knowledge, there is no rigorous understanding of why this happens. This work provides the first theoretical analysis of self-distillation. We focus on fitting a nonlinear function to training data, where the model space is Hilbert space and fitting is subject to $\ell_2$ regularization in this function space. We show that self-distillation iterations modify regularization by progressively limiting the number of basis functions that can be used to represent the solution. This implies (as we also verify empirically) that while a few rounds of self-distillation may reduce over-fitting, further rounds may lead to under-fitting and thus worse performance.

## 1 Introduction

**Knowledge Distillation.** Knowledge distillation was introduced in the deep learning setting [13] as a method for transferring knowledge from one architecture (teacher) to another (student), with the student model often being smaller (see also [5] for earlier ideas). This is achieved by training the student model using the output probability distribution of the teacher model in addition to original labels. The student model benefits from this "dark knowledge" (extra information in soft predictions) and often performs better than if it was trained on the actual labels. Various extensions of this approach have been recently proposed, where instead of output predictions, the student tries to match other statistics from the teacher model such as intermediate feature representations [27], Jacobian matrices [31], distributions [15], Gram matrices [37]. Additional developments on knowledge distillation include its extensions to Bayesian settings [17, 34], uncertainty preservation [33], reinforcement learning [14, 32, 9], online distillation [19], zero-shot learning [24], multi-step knowledge distillation [23], tackling catastrophic forgetting [20], transfer of relational knowledge [25], adversarial distillation [35]. Recently [26] analyzed why the student model is able to mimic teacher model in knowledge distillation and [21] presented a statistical perspective on distillation.

**Self-Distillation.** The special case when the teacher and student architectures are identical is called *self-distillation*. The idea is to feed in predictions of the trained model as new target values for retraining (and iterate this loop possibly a few times). It has been consistently observed that the self-distilled often achieves higher accuracy on held out data [8, 36, 2]. Why this happens, however, has been a mystery: the self-distillation dynamics does not receive any new information about the

task and solely evolves by looping over training. There have been some recent attempts to understand the mysteries around distillation. [11] have empirically observed that the dark knowledge transferred by the teacher is localized mainly in higher layers and does not affect early (feature extraction) layers much. [8] interprets dark knowledge as importance weighting. [6] shows that early-stopping is crucial for reaching dark-knowledge of self-distillation. [1] empirically studies how inductive biases are transferred through distillation. Ideas similar to self-distillation have been used in areas besides modern machine learning but with different names such diffusion and boosting in both the statistics and image processing communities [22].

**Contributions.** Despite interesting developments, why distillation can improve generalization remains elusive. To the best of our knowledge, there is no rigorous understanding of this phenomenon. This work provides a *theoretical analysis of self-distillation*. While originally observed in deep learning, we show that this is a more **profound phenomenon** that can occur even in classical regression settings, where we fit a nonlinear function to training data with models belonging to a Hilbert space and using $\ell_2$ regularization in this function space. In this setting we show that the **self-distillation iterations progressively limit the number of basis functions used to represent the solution**. This implies (as we also verify empirically) that while **a few rounds of self-distillation may reduce over-fitting, further rounds may lead to under-fitting and thus worse performance**. More precisely, we show that self-distillation results in a non-conventional power iteration where the linear operation changes dynamically; each step depends intricately on the results of earlier linear operations via a nonlinear recurrence. We also prove that using **lower training error across distillation steps generally improves the sparsity effect**, and specifically we provide a closed form bound on the sparsity level as the training error goes to zero. Finally, we discuss how our regularization results can be translated into **generalization bounds**.

**Organization.** In Section 2 we setup a variational formulation of nonlinear regression and discuss the existence of non-trivial solutions. Section 3 formalizes self-distillation in our setting. It then shows self-distillation iterations at some point collapse the solution. It provides a lower bound on the number of distillation rounds before the collapse is reached. In addition, it shows that coefficient of the basis functions initially used to represent the solution gradually progressively become sparser. Finally, we discuss that by having the models operate in the near-interpolation regime one can ultimately achieve higher sparsity level. Section 5 draws connection between our setting and the NTK regime of neural networks. This motivates subsequent experiments on deep neural networks in that section. Full **proofs** for these as well as the **code** for reproducing examples in Sections 4 and results in Section 5 are available in the supplementary appendix.

## 2  Problem Setup

We first introduce some notation. We denote a set by $\mathcal{A}$, a matrix by $\boldsymbol{A}$, a vector by $\boldsymbol{a}$, and a scalar by $a$ or $A$. The $(i,j)$'th component of a matrix is denoted by $\boldsymbol{A}[i,j]$ and the $i$'th component of a vector by $\boldsymbol{a}[i]$. Also $\|\,.\,\|$ refers to $\ell_2$ norm of a vector. We use $\triangleq$ to indicate equal by definition. A linear operator $L$ applied to a function $f$ is shown by $[Lf]$, and when evaluated at point $x$ by $[Lf](x)$. For a positive definite matrix $\boldsymbol{A}$, we use $\kappa$ to refer to its condition number $\kappa \triangleq \frac{d_{\max}}{d_{\min}}$, where $d$'s are eigenvalues of $\boldsymbol{A}$. Consider a finite training set $\mathcal{D} \triangleq \cup_{k=1}^K \{(\boldsymbol{x}_k, y_k)\}$, where $\boldsymbol{x}_k \in \mathcal{X} \subseteq \mathbb{R}^d$ and $y_k \in \mathcal{Y} \subseteq \mathbb{R}$. Consider a space of all admissible functions (as we define later in this section) $\mathcal{F} : \mathcal{X} \to \mathcal{Y}$. The goal is to use this training data to find a function $f^* : \mathcal{X} \to \mathcal{Y}$ that approximates the underlying mapping from $\mathcal{X}$ to $\mathcal{Y}$. We assume the function space $\mathcal{F}$ is rich enough to contain multiple functions that can fit finite training data. Thus, the presence of an adequate inductive bias is essential to guide the training process towards solutions that generalize. We infuse such bias in training via regularization. Specifically, we study regression problems of the form[1] below, where $R : \mathcal{F} \to \mathbb{R}$ is a regularization functional, and $\epsilon > 0$ is a desired loss tolerance.

$$f^* \triangleq \arg\min_{f \in \mathcal{F}} R(f) \quad \text{s.t.} \quad \frac{1}{K} \sum_k \big(f(\boldsymbol{x}_k) - y_k\big)^2 \le \epsilon. \tag{1}$$

We consider regularizers with the following type,

$$R(f) = \int_{\mathcal{X}} \int_{\mathcal{X}} u(\boldsymbol{x}, \boldsymbol{x}^\dagger) f(\boldsymbol{x}) f(\boldsymbol{x}^\dagger) \, d\boldsymbol{x} \, d\boldsymbol{x}^\dagger \,, \tag{2}$$

with $u$ being such that $\forall f \in \mathcal{F}$; $R(f) \geq 0$ with *equality* only when $f(\boldsymbol{x}) = 0$. Without loss of generality, we assume $u$ is symmetric $u(\boldsymbol{x}, \boldsymbol{x}^\dagger) = u(\boldsymbol{x}^\dagger, \boldsymbol{x})$. For a given $u$, the space $\mathcal{F}$ of admissible functions are $f$'s for which the double integral in (2) is bounded. The conditions we imposed on $R(f)$ implies that the operator $L$ defined as $[Lf] \triangleq \int_{\mathcal{X}} u(\boldsymbol{x}, .) f(\boldsymbol{x}) \, d\boldsymbol{x}$ has an empty null space[2]. The symmetry and non-negativity conditions together are called *Mercer's condition* and $u$ is called a kernel. Constructing $R$ via kernel $u$ can cover a wide range of regularization forms including the form $R(f) = \int_{\mathcal{X}} \sum_{j=1}^{J} w_j \big([P_j f](\boldsymbol{x})\big)^2 \, d\boldsymbol{x}$, where $P_j$ is some linear operator (e.g. a differential operator to penalize non-smooth functions as we will see in Section 4), and $w_j \geq 0$ is some weight, for $j = 1, \dots, J$ operators. Plugging $R(f)$ into the objective functional leads to the variational problem,

$$f^* \triangleq \arg\min_{f \in \mathcal{F}} \int_{\mathcal{X}} \int_{\mathcal{X}} u(\boldsymbol{x}, \boldsymbol{x}^\dagger) f(\boldsymbol{x}) f(\boldsymbol{x}^\dagger) d\boldsymbol{x} d\boldsymbol{x}^\dagger \quad \text{s.t.} \quad \frac{1}{K} \sum_k \big(f(\boldsymbol{x}_k) - y_k\big)^2 \leq \epsilon \,. \tag{3}$$

The Karush-Kuhn-Tucker (KKT) condition for this problem yields,

$$f_\lambda^* \triangleq \arg\min_{f \in \mathcal{F}} \frac{\lambda}{K} \sum_k \big(f(\boldsymbol{x}_k) - y_k\big)^2 + \int_{\mathcal{X}} \int_{\mathcal{X}} u(\boldsymbol{x}, \boldsymbol{x}^\dagger) f(\boldsymbol{x}) f(\boldsymbol{x}^\dagger) \, d\boldsymbol{x} \, d\boldsymbol{x}^\dagger \tag{4}$$

$$\text{s.t.} \quad \lambda \geq 0 \quad , \quad \frac{1}{K} \sum_k \big(f(\boldsymbol{x}_k) - y_k\big)^2 \leq \epsilon \quad , \quad \lambda\big(\frac{1}{K} \sum_k \big(f(\boldsymbol{x}_k) - y_k\big)^2 - \epsilon\big) = 0 \,. \tag{5}$$

## 2.1 Existence of Non-Trivial Solutions

Stack training labels into a vector,

$$\boldsymbol{y}_{K \times 1} \triangleq [\, y_1 \,|\, y_2 \,|\, \dots \,|\, y_K \,] \,. \tag{6}$$

It is obvious that when $\frac{1}{K}\|\boldsymbol{y}\|^2 \leq \epsilon$, then $f^*$ has trivial solution $f^*(\boldsymbol{x}) = 0$, which we refer to this case as *collapse* regime. In the sequel, we focus on the more interesting case of $\frac{1}{K}\|\boldsymbol{y}\|^2 > \epsilon$. It is also easy to verify that collapsed solution is tied to $\lambda = 0$,

$$\|\boldsymbol{y}\|^2 > K\epsilon \quad \Leftrightarrow \quad \lambda > 0 \,. \tag{7}$$

Thus by taking any $\lambda > 0$ that satisfies $\frac{1}{K} \sum_k \big(f_\lambda^*(\boldsymbol{x}_k) - y_k\big)^2 - \epsilon = 0$, the following form $f_\lambda^*$ is an optimal solution to the problem (3), i.e. $f^* = f_\lambda^*$.

$$f_\lambda^* = \arg\min_{f \in \mathcal{F}} \frac{\lambda}{K} \sum_k \big(f(\boldsymbol{x}_k) - y_k\big)^2 + \int_{\mathcal{X}} \int_{\mathcal{X}} u(\boldsymbol{x}, \boldsymbol{x}^\dagger) f(\boldsymbol{x}) f(\boldsymbol{x}^\dagger) \, d\boldsymbol{x} \, d\boldsymbol{x}^\dagger \,. \tag{8}$$

## 2.2 Closed Form of $f^*$

In this section we present a closed form expression for (8). Since we are considering $\lambda > 0$, without loss of generality, we can divide the objective function in (8) by $\lambda$ and let $c \triangleq 1/\lambda$; obviously $c > 0$.

$$f^* = \arg\min_{f \in \mathcal{F}} \frac{1}{K} \sum_k \big(f(\boldsymbol{x}_k) - y_k\big)^2 + c \int_{\mathcal{X}} \int_{\mathcal{X}} u(\boldsymbol{x}, \boldsymbol{x}^\dagger) f(\boldsymbol{x}) f(\boldsymbol{x}^\dagger) \, d\boldsymbol{x} \, d\boldsymbol{x}^\dagger \,. \tag{9}$$

The variational problem (9) has appeared in machine learning context extensively [10]. It has a known solution form, due to representer theorem [29], which we will present here in a proposition. However, we first need to introduce some definitions. Let $g(\boldsymbol{x}, \boldsymbol{t})$ be a function such that $\int_{\mathcal{X}} u(\boldsymbol{x}, \boldsymbol{x}^\dagger) \, g(\boldsymbol{x}^\dagger, \boldsymbol{t}) \, d\boldsymbol{x}^\dagger = \delta(\boldsymbol{x} - \boldsymbol{t})$, where $\delta(\boldsymbol{x})$ is Dirac delta. Such $g$ is called the *Green's function*[3] of the linear operator $L$, with $L$ being $[Lf](\boldsymbol{x}) \triangleq \int_{\mathcal{X}} u(\boldsymbol{x}, \boldsymbol{x}^\dagger) \, f(\boldsymbol{x}^\dagger) \, d\boldsymbol{x}^\dagger$. Let the matrix $\boldsymbol{G}_{K \times K}$ and the vector $\boldsymbol{g}_{\boldsymbol{x} \, K \times 1}$ be defined as,

$$\boldsymbol{G}[j, k] \triangleq \frac{1}{K} g(\boldsymbol{x}_j, \boldsymbol{x}_k) \quad , \quad \boldsymbol{g}_{\boldsymbol{x}}[k] \triangleq \frac{1}{K} g(\boldsymbol{x}, \boldsymbol{x}_k) \,. \tag{10}$$

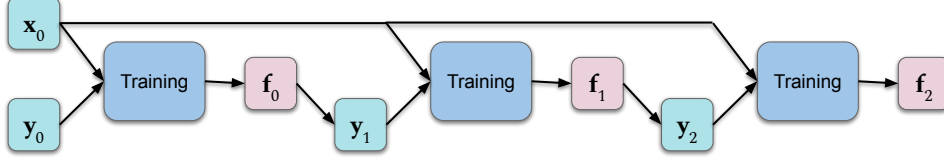

Figure 1: Schematic illustration of the self-distillation process for two iterations.

**Proposition 1** *The variational problem (9) has a solution of the form,*

$$f^*(\boldsymbol{x}) \;=\; \boldsymbol{g}_{\boldsymbol{x}}^T (c\boldsymbol{I} + \boldsymbol{G})^{-1}\boldsymbol{y}\,. \tag{11}$$

Notice that the matrix $\boldsymbol{G}$ is *positive definite*[4]. Since by definition $c > 0$, the inverse of the matrix $c\boldsymbol{I} + \boldsymbol{G}$ is well-defined. Also, because $\boldsymbol{G}$ is positive definite, it can be diagonalized as $\boldsymbol{G} = \boldsymbol{V}^T \boldsymbol{D} \boldsymbol{V}$, where the diagonal matrix $\boldsymbol{D}$ contains the eigenvalues of $\boldsymbol{G}$, denoted as $d_1, \ldots, d_K$ that are strictly greater than zero, and the matrix $\boldsymbol{V}$ contains the corresponding eigenvectors.

## 2.3 Bounds on Multiplier $c$

Earlier we showed that any $c > 0$ that is a root of $\frac{1}{K} \sum_k \left( f_c^*(\boldsymbol{x}_k) - y_k \right)^2 - \epsilon = 0$ produces an optimal solution $f^*$ via (9). However, in the settings that we are interested in, we do not know the underlying $c$ or $f^*$ beforehand; we have to relate them the given training data instead. As we will see later in Proposition 3, knowledge of $c$ allows us to resolve the recurrence on $\boldsymbol{y}$ created by self-distillation loop and obtain an explicit bound $\|\boldsymbol{y}\|$ at each distillation round. Unfortunately finding $c$ in closed form by seeking roots of $\frac{1}{K} \sum_k \left( f_c^*(\boldsymbol{x}_k) - y_k \right)^2 - \epsilon = 0$ w.r.t. $c$ is impossible, due to the nonlinear dependency of $f$ on $c$ caused by matrix inversion; see (11). However, we can still provide bounds on the value of $c$ as shown in this section. Throughout the analysis, it is sometimes easier to work with rotated labels $\boldsymbol{V}\boldsymbol{y}$. Thus we define $\boldsymbol{z} \triangleq \boldsymbol{V}\boldsymbol{y}$. Note that any result on $\boldsymbol{z}$ can be easily converted back via $\boldsymbol{y} = \boldsymbol{V}^T \boldsymbol{z}$, as $\boldsymbol{V}$ is an orthogonal matrix. Trivially $\|\boldsymbol{z}\| = \|\boldsymbol{y}\|$. Our first step is to present a simplified form for the error term $\frac{1}{K} \sum_k \left( f^*(\boldsymbol{x}_k) - y_k \right)^2$ using the following proposition.

**Proposition 2** *The following identity holds* $\frac{1}{K} \sum_k \left( f^*(\boldsymbol{x}_k) - y_k \right)^2 \;=\; \frac{1}{K} \sum_k (z_k \frac{c}{c+d_k})^2.$

We now proceed to bound the roots of $h(c) \triangleq \frac{1}{K} \sum_k (z_k \frac{c}{c+d_k})^2 - \epsilon$. Since we are considering $\|\boldsymbol{y}\|^2 > K\epsilon$, and thus by (7) $c > 0$, it is easy to construct the following lower and upper bounds on $h$ $\underline{h}(c) \triangleq \frac{1}{K} \sum_k (z_k \frac{c}{c+d_{\max}})^2 - \epsilon$, $\overline{h}(c) \triangleq \frac{1}{K} \sum_k (z_k \frac{c}{c+d_{\min}})^2 - \epsilon$. The roots of $\underline{h}$ and $\overline{h}$, namely $c_1$ and $c_2$, can be easily derived $c_1 = \frac{d_{\max}\sqrt{K\epsilon}}{\|\boldsymbol{z}\| - \sqrt{K\epsilon}}$, $c_2 = \frac{d_{\min}\sqrt{K\epsilon}}{\|\boldsymbol{z}\| - \sqrt{K\epsilon}}$. Since $\underline{h}(c) \le h(c)$, the condition $\underline{h}(c_1) = 0$ implies that $h(c_1) \ge 0$. Similarly, since $h(x) \le \overline{h}(c)$, the condition $\overline{h}(c_2) = 0$ implies that $h(c_2) \le 0$. By the intermediate value theorem, due to continuity of $f$ and the fact that $\|\boldsymbol{z}\| = \|\boldsymbol{y}\| > \sqrt{K\epsilon}$ (non-collapse condition), there is a point $c$ between $c_1$ and $c_2$ at which $h(c) = 0$,

$$\frac{d_{\min}\sqrt{K\epsilon}}{\|\boldsymbol{z}\| - \sqrt{K\epsilon}} \le c \le \frac{d_{\max}\sqrt{K\epsilon}}{\|\boldsymbol{z}\| - \sqrt{K\epsilon}}. \tag{12}$$

## 3 Self-Distillation

Denote the prediction vector over the training data $\boldsymbol{x}_1, \ldots \boldsymbol{x}_K$ as $\boldsymbol{f}_{K \times 1} \triangleq \left[ f^*(\boldsymbol{x}_1) \,|\, \ldots \,|\, f^*(\boldsymbol{x}_K) \right]^T = \boldsymbol{V}^T \boldsymbol{D}(c\boldsymbol{I} + \boldsymbol{D})^{-1}\boldsymbol{V}\boldsymbol{y}$. Self-distillation treats this prediction as target labels for a new round of training, and repeats this process as shown in Figure 1. With abuse of notation, denote the $t$'th round of distillation by subscript $t$. We refer to the

standard training (no self-distillation yet) by the step $t = 0$. Thus the standard training step has the form $\boldsymbol{f}_0 = \boldsymbol{V}^T \boldsymbol{D}(c_0 \boldsymbol{I} + \boldsymbol{D})^{-1} \boldsymbol{V} \boldsymbol{y}_0$, where $\boldsymbol{y}_0$ is the ground truth labels as defined in (6). Letting $\boldsymbol{y}_1 \triangleq \boldsymbol{f}_0$, we achieve the next model by applying the first round of self-distillation $\boldsymbol{f}_1 = \boldsymbol{V}^T \boldsymbol{D}(c_1 \boldsymbol{I} + \boldsymbol{D})^{-1} \boldsymbol{V} \boldsymbol{y}_1$. In general, for any $t \geq 1$ we have the following recurrence,

$$\forall t \geq 1 \,;\; \boldsymbol{y}_t = \boldsymbol{V}^T \boldsymbol{A}_{t-1} \boldsymbol{V} \boldsymbol{y}_{t-1} \quad , \quad \forall t \geq 0 \,;\; \boldsymbol{A}_{t\,K \times K} \triangleq \boldsymbol{D}(c_t \boldsymbol{I} + \boldsymbol{D})^{-1} \,. \tag{13}$$

Note that $\boldsymbol{A}_t$ is also a diagonal matrix. Furthermore, since at the $t$'th distillation step, everything is the same as the initial step except the training labels, we can use Proposition 1 to express $f_t(\boldsymbol{x})$ as,

$$f_t^*(\boldsymbol{x}) = \boldsymbol{g}_{\boldsymbol{x}}^T (c_t \boldsymbol{I} + \boldsymbol{G})^{-1} \boldsymbol{y}_t = \boldsymbol{g}_{\boldsymbol{x}}^T \boldsymbol{V}^T \boldsymbol{D}^{-1} (\Pi_{i=0}^t \boldsymbol{A}_t) \boldsymbol{V} \boldsymbol{y}_0 \,. \tag{14}$$

Observe that the only dynamic component in the expression of $f_t^*$ is $\Pi_{i=0}^t \boldsymbol{A}_i$. In the following, we show how $\Pi_{i=0}^t \boldsymbol{A}_i$ evolves over time. Specifically, we show self-distillation progressively sparsifies the matrix $\Pi_{i=0}^t \boldsymbol{A}_i$ at a provided rate. Also recall from Section 2.1 that *in each step of self-distillation* we require $\|\boldsymbol{y}_t\| > \sqrt{K}\epsilon$. If this condition breaks, the solution *collapses* to zero function and subsequent rounds of self-distillation keep producing $f^*(\boldsymbol{x}) = 0$. In this section we present a lower bound on number of iterations $t$ guaranteeing all intermediate problems satisfy $\|\boldsymbol{y}_t\| > \sqrt{K}\epsilon$.

### 3.1 Unfolding the Recurrence

Our goal here is to understand how $\|\boldsymbol{y}_t\|$ evolves in $t$. By combining the equations in (13) we obtain $\boldsymbol{y}_t = \boldsymbol{V}^T \boldsymbol{D}(c_{t-1} \boldsymbol{I} + \boldsymbol{D})^{-1} \boldsymbol{V} \boldsymbol{y}_{t-1}$. By multiplying both sides from the left by $\boldsymbol{V}$ we get $\boldsymbol{V} \boldsymbol{y}_t = \boldsymbol{V} \boldsymbol{V}^T \boldsymbol{D}(c_{t-1} \boldsymbol{I} + \boldsymbol{D})^{-1} \boldsymbol{V} \boldsymbol{y}_{t-1}$, which is equivalent to,

$$\boldsymbol{z}_t = \boldsymbol{D}(c_{t-1} \boldsymbol{I} + \boldsymbol{D})^{-1} \boldsymbol{z}_{t-1} \equiv \frac{1}{\sqrt{K}\epsilon} \boldsymbol{z}_t = \boldsymbol{D}(c_{t-1} \boldsymbol{I} + \boldsymbol{D})^{-1} \frac{1}{\sqrt{K}\epsilon} \boldsymbol{z}_{t-1} \,. \tag{15}$$

Also we can use the bounds on $c$ from (12) at any arbitrary $t \geq 0$ and thus write ,

$$\forall t \geq 0 \,;\; \|\boldsymbol{z}_t\| > \sqrt{K}\epsilon \;\Rightarrow\; \frac{d_{\min}\sqrt{K}\epsilon}{\|\boldsymbol{z}_t\| - \sqrt{K}\epsilon} \leq c_t \leq \frac{d_{\max}\sqrt{K}\epsilon}{\|\boldsymbol{z}_t\| - \sqrt{K}\epsilon} \tag{16}$$

By combining RHS of (15) and (16) we obtain a recurrence solely in $\boldsymbol{z}$ as shown below, where $d_{\min} \leq \alpha_t \leq d_{\max}$.

$$\boldsymbol{z}_t = \boldsymbol{D}\left(\frac{\alpha_t \sqrt{K}\epsilon}{\|\boldsymbol{z}_{t-1}\| - \sqrt{K}\epsilon} \boldsymbol{I} + \boldsymbol{D}\right)^{-1} \boldsymbol{z}_{t-1} \,. \tag{17}$$

We now establish a lower bound on the value of $\|\boldsymbol{z}_t\|$.

**Proposition 3** *For any $t \geq 0$, if $\|\boldsymbol{z}_i\| > \sqrt{K}\epsilon$ for $i = 0, \ldots, t$, then $\|\boldsymbol{z}_t\| \geq a^t(\kappa)\|\boldsymbol{z}_0\| - \sqrt{K}\epsilon\, b(\kappa) \frac{a^t(\kappa)-1}{a(\kappa)-1}$, where $a(x) \triangleq \frac{(r_0-1)^2 + x(2r_0-1)}{(r_0-1+x)^2}$, $b(x) \triangleq \frac{r_0^2 x}{(r_0-1+x)^2}$, $r_0 \triangleq \frac{1}{\sqrt{K}\epsilon}\|\boldsymbol{z}_0\|$, $\kappa \triangleq \frac{d_{\max}}{d_{\min}}$.*

### 3.2 Guaranteed Number of Self-Distillation Rounds

By looking at the LHS of (15) it is not hard to see the value of $\|\boldsymbol{z}_t\|$ is ***decreasing*** in $t$. That is because $c_t$[5] as well as elements of the diagonal matrix $\boldsymbol{D}$ are strictly positive. Hence $\boldsymbol{D}(c_{t-1} \boldsymbol{I} + \boldsymbol{D})^{-1}$ shrinks the magnitude of $\boldsymbol{z}_{t-1}$ in each iteration. Thus, starting from $\|\boldsymbol{z}_0\| > \sqrt{K}\epsilon$, as $\|\boldsymbol{z}_t\|$ decreases, at some point it falls below the value $\sqrt{K}\epsilon$ and thus the solution collapses. We now want to find out after how many rounds $t$, the solution collapse happens. Finding the exact such $t$, is difficult, but by setting a lower bound of $\|\boldsymbol{z}_t\|$ to $\sqrt{K}\epsilon$ and solving that in $t$, calling the solution $\underline{t}$, we can guarantee realization of at least $\underline{t}$ rounds where the value of $\|\boldsymbol{z}_{\underline{t}}\|$ remains above $\sqrt{K}\epsilon$. We can use the lower bound we developed in Proposition 3 in order to find such $\underline{t}$. This is shown in the following proposition. Note that when we are in near-interpolation regime, i.e. $\epsilon \to 0$, the form of $\underline{t}$ simplifies: $\underline{t} \approx \frac{\|\boldsymbol{y}_0\|}{\kappa \sqrt{K}\epsilon}$.

**Proposition 4** *Starting from $\|\boldsymbol{y}_0\| > \sqrt{K}\epsilon$, meaningful (non-collapsing solution) self-distillation is possible at least for $\underline{t}$ rounds,*

$$\underline{t} \triangleq \frac{\frac{\|\boldsymbol{y}_0\|}{\sqrt{K}\epsilon} - 1}{\kappa} \,. \tag{18}$$

## 3.3 Evolution of Basis

Recall from (14) that the learned function after $t$ rounds of self-distillation has the form $f_t^*(\boldsymbol{x}) = \boldsymbol{g}_{\boldsymbol{x}}^T \boldsymbol{V}^T \boldsymbol{D}^{-1}(\Pi_{i=0}^t \boldsymbol{A}_t) \boldsymbol{V} \boldsymbol{y}_0$. The only time-dependent part is thus the following *diagonal* matrix $\boldsymbol{B}_t$ defined in (19). In this section we show how $\boldsymbol{B}_t$ evolves over time. Specifically, we claim that the matrix $\boldsymbol{B}_t$ becomes progressively sparser as $t$ increases.

$$\boldsymbol{B}_t \triangleq \Pi_{i=0}^t \boldsymbol{A}_t \,. \tag{19}$$

**Theorem 5** *Suppose $\|\boldsymbol{y}_0\| > \sqrt{K}\epsilon$ and $t \leq \frac{\|\boldsymbol{y}_0\|}{\kappa\sqrt{K}\epsilon} - \frac{1}{\kappa}$. Then for any pair of diagonals of $\boldsymbol{D}$, namely $d_j$ and $d_k$, with the condition that $d_k > d_j$, the following inequality holds.*

$$\frac{\boldsymbol{B}_{t-1}[k,k]}{\boldsymbol{B}_{t-1}[j,j]} \geq \left( \frac{\frac{\|\boldsymbol{y}_0\|}{\sqrt{K}\epsilon} - 1 + \frac{d_{\min}}{d_j}}{\frac{\|\boldsymbol{y}_0\|}{\sqrt{K}\epsilon} - 1 + \frac{d_{\min}}{d_k}} \right)^t \,. \tag{20}$$

The above theorem suggests that, as $t$ increases, the smaller elements of $\boldsymbol{B}_{t-1}$ shrink faster and at some point become negligible compared to larger ones. That is because in (20) we have assumed $d_k > d_j$, and thus the r.h.s. expression in the parentheses is strictly greater than 1. The latter implies that $\frac{\boldsymbol{B}_{t-1}[k,k]}{\boldsymbol{B}_{t-1}[j,j]}$ is increasing in $t$. Observe that if one was able to push $t \to \infty$, then only one entry of $\boldsymbol{B}_t$ (the one corresponding to $d_{\max}$) would remains significant relative to others. Thus, self-distillation process *progressively sparsifies* $\boldsymbol{B}_t$. This sparsification affects the expressiveness of the regression solution $f_t^*(\boldsymbol{x})$. To see that, use the definition of $f_t^*(\boldsymbol{x})$ from (14) to express it in the form (21), where we gave a name to the rotated and scaled basis $\boldsymbol{p}_{\boldsymbol{x}} \triangleq \boldsymbol{D}^{-1}\boldsymbol{V}\boldsymbol{g}_{\boldsymbol{x}}$ and rotated vector $\boldsymbol{z}_0 \triangleq \boldsymbol{V}\boldsymbol{y}_0$. The solution $f_t^*$ is essentially represented by a weighted sum of the basis functions (the components of $\boldsymbol{p}_{\boldsymbol{x}}$). Thus, the number of significant diagonal entries of $\boldsymbol{B}_t$ determines the *effective number of basis functions* used to represent the solution.

$$f_t^*(\boldsymbol{x}) = \boldsymbol{g}_{\boldsymbol{x}}^T \boldsymbol{V}^T \boldsymbol{D}^{-1} \boldsymbol{B}_t \boldsymbol{V} \boldsymbol{y}_0 = \boldsymbol{p}_{\boldsymbol{x}}^T \boldsymbol{B}_t \boldsymbol{z}_0 \,. \tag{21}$$

## 3.4 Self-Distillation versus Early Stopping

Broadly speaking, early stopping can be interpreted as any procedure that cuts convergence short of the optimal solution. Examples include reducing the number of iterations of the numerical optimizer (e.g. SGD), or increasing the loss tolerance threshold $\epsilon$. The former is not applicable to our setting, as our analysis is independent of function parametrization and its numerical optimization. We consider the second definition. This form of early stopping also has a regularization effect; by increasing $\epsilon$ in (1) the feasible set expands and thus it is possible to find functions with lower $R(f)$. However, we show here that the induced regularization is not equivalent to that of self-distillation. In fact, one can say that early-stopping does the *opposite* of sparsification, as we show below. The learned function via loss-based early stopping in our notation can be expressed as $f_0^*$ (single training, no self-distillation) with a larger error tolerance $\epsilon$,

$$f_0^*(\boldsymbol{x}) = \boldsymbol{p}_{\boldsymbol{x}}^T \boldsymbol{B}_0 \boldsymbol{z}_0 = \boldsymbol{p}_{\boldsymbol{x}}^T \boldsymbol{D}(c_0 \boldsymbol{I} + \boldsymbol{D})^{-1} \boldsymbol{z}_0 \,. \tag{22}$$

The effect of larger $\epsilon$ on the value of $c_0$ is shown in (12). However, since $c_0$ is just a scalar value applied to matrices, it does not provide any lever to increase the sparsity of $\boldsymbol{D}$. We now elaborate on the latter claim a bit more. Observe that, on the one hand, when $c_0$ is large, then $\boldsymbol{D}(c_0 \boldsymbol{I} + \boldsymbol{D})^{-1} \approx \frac{1}{c_0}\boldsymbol{D}$, which essentially uses $\boldsymbol{D}$ and does not sparsify it further. On the other hand, if $c_0$ is small then $\boldsymbol{D}(c_0 \boldsymbol{I} + \boldsymbol{D})^{-1} \approx \boldsymbol{I}$, which is the densest possible diagonal matrix. Thus, at best, early stopping maintains the original sparsity pattern of $\boldsymbol{D}$ and otherwise makes it even denser.

## 3.5 Advantage of Near Interpolation Regime

As discussed in Section (3.3), one can think of $\frac{\boldsymbol{B}_{t-1}[k,k]}{\boldsymbol{B}_{t-1}[j,j]}$ as a sparsity measure (the larger, the sparser). Thus, we define a *sparsity index* based on the lower bound we developed for $\frac{\boldsymbol{B}_{t-1}[k,k]}{\boldsymbol{B}_{t-1}[j,j]}$ in

Proposition 5. In fact, by finding the lowest value of the bound across elements all elements satisfying $d_k > d_j$ and further assuming $d_1 < d_2 < \cdots < d_K$, we can ensure at least the sparsity level of,

$$S_{\boldsymbol{B}_{t-1}} \triangleq \min_{k \in \{1,2,\ldots,K-1\}} \left( \frac{\frac{\|\boldsymbol{y}_0\|}{\sqrt{K}\,\epsilon} - 1 + \frac{d_{\min}}{d_k}}{\frac{\|\boldsymbol{y}_0\|}{\sqrt{K}\,\epsilon} - 1 + \frac{d_{\min}}{d_{k+1}}} \right)^t. \tag{23}$$

One may wonder what is the highest sparsity $S$ that self-distillation can attain. Since $\|\boldsymbol{y}_0\| > \sqrt{K}\,\epsilon$ and $d_{k+1} > d_k$, the term inside parentheses in (23) is strictly greater than 1 and thus $S$ increases in $t$. However, the largest $t$ we can guarantee before a solution collapse (see Proposition 4) is $\underline{t} = \frac{\|\boldsymbol{y}_0\|}{\kappa\sqrt{K}\,\epsilon} - \frac{1}{\kappa}$. By plugging this $\underline{t}$ into the definition of $S$ (23) we eliminate $t$ and obtain the largest sparsity index as shown in (24). In the next theorem, we show $S_{\boldsymbol{B}_{t-1}}$ always improves as $\epsilon$ gets smaller. Thus, if high sparsity is desired, one can set $\epsilon$ as small as possible. One should however note that the value of $\epsilon$ cannot be *identically zero*, i.e. exact interpolation regime, because then $\boldsymbol{f}_0 = \boldsymbol{y}_0$, and since $\boldsymbol{y}_1 = \boldsymbol{f}_0$, self-distillation process keeps producing the same model in each round.

$$S_{\boldsymbol{B}_{\underline{t}-1}} = \min_{k \in \{1,2,\ldots,K-1\}} \left( \frac{\frac{\|\boldsymbol{y}_0\|}{\sqrt{K}\,\epsilon} - 1 + \frac{d_{\min}}{d_k}}{\frac{\|\boldsymbol{y}_0\|}{\sqrt{K}\,\epsilon} - 1 + \frac{d_{\min}}{d_{k+1}}} \right)^{\frac{\|\boldsymbol{y}_0\|}{\kappa\sqrt{K}\,\epsilon} - \frac{1}{\kappa}}. \tag{24}$$

**Theorem 6** *Suppose $\|\boldsymbol{y}_0\| > \sqrt{K}\,\epsilon$. Then the sparsity index $S_{\boldsymbol{B}_{\underline{t}-1}}$ (where $\underline{t} = \frac{\|\boldsymbol{y}_0\|}{\kappa\sqrt{K}\,\epsilon} - \frac{1}{\kappa}$ is number of guaranteed self-distillation steps before solution collapse) decreases in $\epsilon$, i.e. lower $\epsilon$ yields higher sparsity. Furthermore at the limit $\epsilon \to 0$, the sparsity index $\lim_{\epsilon \to 0} S_{\boldsymbol{B}_{\underline{t}-1}} = e^{\frac{d_{\min}}{\kappa} \min_{k \in \{1,2,\ldots,K-1\}} \left( \frac{1}{d_k} - \frac{1}{d_{k+1}} \right)}$.*

## 3.6 Multiclass Extension

We can formulate multiclass classification, by regressing to a one-hot encoding. Specifically, a problem with $Q$ classes can be modeled by $Q$ output functions $f_1, \ldots, f_Q$. An easy extension of our analysis to this multiclass setting is to require the functions $f_1, \ldots, f_Q$ be smooth by applying the same regularization $R$ to each and then adding up these regularization terms. This way, the optimal function for each output unit can be solved for each $q = 1, \ldots, Q$ as $f_q^* \triangleq \arg\min_{f_q \in \mathcal{F}} \frac{1}{K} \sum_k (f_q(\boldsymbol{x}_k) - y_{q_k})^2 + c_q R(f_q)$.

## 3.7 Generalization Bounds

Our result can be easily translated into generalization guarantees. Recall from (14) that the regression solution after $t$ rounds of self-distillation has the form $f_t^*(\boldsymbol{x}) = \boldsymbol{g}_{\boldsymbol{x}}^T \boldsymbol{V}^T \boldsymbol{D}^{-1} (\Pi_{i=0}^t \boldsymbol{A}_t) \boldsymbol{V} \boldsymbol{y}_0$. We can show that (proof in Appendix B), there exists a positive definite kernel $g^\dagger(.,.)$ that performing standard kernel ridge regression with it over the same training data $\cup_{k=1}^K \{(\boldsymbol{x}_k, y_k)\}$ yields the function $f^\dagger$ such that $f^\dagger = f_t^*$. Furthermore, we can show that the spectrum of the Gram matrix $\boldsymbol{G}^\dagger[j,k] \triangleq \frac{1}{K} g^\dagger(\boldsymbol{x}_j, \boldsymbol{x}_k)$ in the latter kernel regression problem relates to spectrum of $\boldsymbol{G}$ via,

$$d_k^\dagger = c_0 \frac{1}{\frac{\Pi_{i=0}^t (d_k + c_i)}{d_k^{t+1}} - 1}. \tag{25}$$

The identity (25) enables us to leverage existing generalization bounds for standard kernel ridge regression. These results often only need the spectrum of the Gram matrix. For example, Lemma 22 in [4] shows the Rademacher complexity of the kernel class is proportional to $\sqrt{\text{tr}(\boldsymbol{G}^\dagger)} = \sqrt{\sum_{k=1}^K d_k^\dagger}$ and then Theorem 8 of [4] translates that Rademacher complexity into a generalization bound. Note that $\frac{\Pi_{i=0}^t (d_k + c_i)}{d_k^{t+1}}$ increases in $t$, which implies $d_k^\dagger$ and consequently $\sqrt{\text{tr}(\boldsymbol{G}^\dagger)}$ decreases in $t$.

A more refined bound in terms of the tail behavior of the eigenvalues $d_k^\dagger$ (to better exploit the sparsity pattern) is the Corollary 6.7 of [3] which provides a generalization bound that is affine in the form

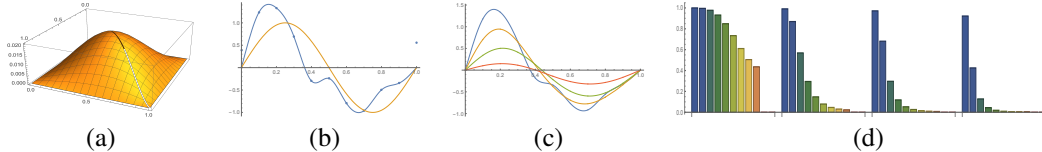

(a)       (b)       (c)       (d)

Figure 2: (a,b,c) Example with $R(f)(x) \triangleq \int_0^1 \left( \frac{d^2}{dx^2} f(x) \right)^2 dx$. (a) Green's function associated with the kernel of $R$. (b) Noisy training samples (blue dots) from underlying function (orange) $y = \sin(2\pi x)$. Fitting without regularization leads to overfitting (blue curve). (c) Four rounds of self-distillation (blue, orange, green, red) with $\epsilon = 0.04$. (d) Evolution of diagonal entries of (diagonal matrix) $\boldsymbol{B}_t$ from (19) at distillation rounds $t = 0$ (left most) to $t = 3$ (right most). The number of training points is $K = 11$, so $\boldsymbol{B}_t$ which is $K \times K$ diagonal matrix has 11 entries on its diagonal, each corresponding to one of the bars in the chart.

$\min_{k \in \{0,1,\dots,K\}} \left( \frac{k}{K} + \sqrt{\frac{1}{K} \sum_{j=k+1}^{K} d_j^\dagger} \right)$, where the eigenvalues $d_k^\dagger$ for $k = 1, \dots, K$, are sorted in non-increasing order .

## 4 Illustrative Example

Let $\mathcal{F}$ be the space of twice differentiable functions that map $[0, 1]$ to $\mathbb{R}$ as $\mathcal{F} \triangleq \{f \mid f : [0, 1] \to \mathbb{R}\}$. Define the linear operator $P : \mathcal{F} \to \mathcal{F}$ as $[Pf](x) \triangleq \frac{d^2}{dx^2} f(x)$ subject to boundary conditions $f(0) = f(1) = f''(0) = f''(1) = 0$. The associated regularization functional becomes $R(f) \triangleq \int_0^1 \left( \frac{d^2}{dx^2} f(x) \right)^2 dx$. Observe that this regularizer encourages smoother $f$ by penalizing the second order derivative of the function. The Green's function of the operator associated with the kernel of $R$ subject to the listed boundary conditions is a spline $g(x, x^\dagger) = \frac{1}{6} \max \left( (x - x^\dagger)^3, 0 \right) - \frac{1}{6} x (1 - x^\dagger)(x^2 - 2x^\dagger + x^{\dagger 2})$ [28] (see Figure 2-a). Now consider training points $(x_k, y_k)$ sampled from the function $y = \sin(2\pi x)$. Let $x_k$ be evenly spaced in the interval $[0, 1]$ with steps of 0.1, and $y_k = x_k + \eta$ where $\eta$ is a zero-mean normal random variable with $\sigma = 0.5$ (Figure 2-b). As shown in Figure 2-c, the regularization induced by self-distillation initially improves the quality of the fit, but after that point additional rounds of self-distillation over-regularize and lead to underfitting. We also computed the diagonal matrix $\boldsymbol{B}_t$ (see (19) for definition) at each self-distillation round $t$, for $t = 0, \dots, 3$ (after that, the solution collapses). Recall from (21) that the entries of this matrix can be thought of as the coefficients of basis functions used to represent the solution. As predicted by our analysis, self-distillation regularizes the solution by sparsifying these coefficients. This is evident in Figure 2-b where smaller coefficients shrink faster.

## 5 Experiments

In our experiments, we aim to empirically evaluate our theoretical analysis in the setting of deep networks. Although our theoretical results apply to Hilbert space rather than deep neural networks, recent findings show that at least very wide neural networks (NTK Regime) can be viewed as a reproducing kernel Hilbert space [16].

We adopt a clear and simple setup that is easy to reproduce (see the provided code) and also light-weight enough to run more then 10 rounds of self-distillation. Readers interested in stronger baselines are referred to [8, 36, 2]. However, these works are limited to one or two rounds of self-distillation. The ability to run self-distillation for a larger number of rounds allows us to demonstrate the eventual decline of the test performance. To the best of our knowledge, this is the first time that the performance *decline regime* is observed. The initial improvement and later continuous decline is consistent with our theory, which shows rounds of self-distillation continuously amplify the regularization effect. While initially this may benefit generalization, at some point the excessive regularization leads to under-fitting.

We use Resnet [12] and VGG [30] neural architectures and train them on CIFAR-10 and CIFAR-100 datasets [18]. Training details and additional results are left to the appendix. Each curve in the plots corresponds to 10 runs from randomly initialized weights, where each run is a chain of self-distillation

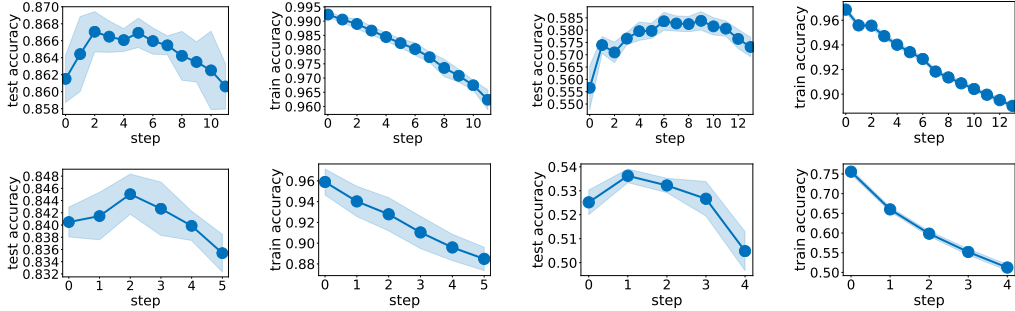

Figure 3: (Top/Bottom): Accuracy of self-distillation steps using Resnet with $\ell_2$/cross-entropy loss. (Left Two Plots / Right Two Plots) test and train accuracy on CIFAR-10/CIFAR-100.

steps indicated in the $x$-axis. In the plots, a point represents the average and the envelope around it reflects standard deviation. Any training accuracy reported here is based on assessing the model $f_t$ at the $t$'th self-distillation round on the ***original*** training labels $\boldsymbol{y}_0$. WE first train the neural network using $\ell_2$ loss. The error is defined as the difference between predictions (softmax over the logits) and the target labels. These results are in concordance with a regularization viewpoint of self-distillation. The theory suggests that self-distillation progressively amplifies the underlying regularization effect. As such, we expect the training accuracy (over $\boldsymbol{y}_0$) to drop in each self-distillation round. Test accuracy may go up if training can benefit from amplified regularization. However, from the theory we expect the test accuracy to go down at some point due to over regularization and thus underfitting. Both of these phenomena are observed in four left plots Figure 3. Although, our theory only applies to $\ell_2$, loss, we empirically observed similar phenomena for cross entropy as shown in four right plots in Figure 3. We have included additional plots in the *appendix* showing the performance of $\ell_2$ distillation on CIFAR-100 using VGG network (hence concluding that the theory and empirical findings are not dependent to a specific structure and apply to architectures beyond Resnet). In the appendix we have also shown that self-distillation and early-stopping have different regularization effects.

## 6 Conclusion

In this work, we presented a rigorous analysis of self-distillation for ridge regression in a Hilbert space of functions. We showed that self-distillation iterations in the setting we studied cannot continue indefinitely; at some point the solution collapses to zero. We provided a lower bound on the number of meaningful (non-collapsed) distillation iterations. In addition, we proved that self-distillation acts as a regularizer that progressively employs fewer basis functions for representing the solution. We discussed the difference in regularization effect induced by self-distillation against early stopping. We also showed that operating in near-interpolation regime facilitates the regularization effect. We discussed how our regression setting resembles the NTK view of wide neural networks, and thus may provide some insight into how self-distillation works in deep learning. We hope that our work can be used as a stepping stone to broader settings. In particular, studying cross-entropy loss as well as other forms of regularization are interesting directions for further research.

### Broader Impact

We believe that this paper is categorized as fundamental and theoretical research and is not targeted to any specific application area. The insights and theory developed here may inspire novel algorithms and more investigations in knowledge distillation and more generally in neural network regularization and generalization. Consequently this may lead to better training algorithms with lower training time, computational cost, or energy consumption. The research presented here can be used for many different application areas and a particular use may have both positive or negative implications. Though, we are not aware of any immediate short term negative impact of this research.

### Acknowledgement

We would like to thank colleagues at Google Research for their feedback and comments: Moshe Dubiner, Pierre Foret, Sergey Ioffe, Yiding Jiang, Alan MacKey, Matt Streeter, and Andrey Zhmoginov.

## Footnotes

[1]Our choice of setting up learning as a constrained optimization rather than unconstrained form $\frac{1}{K} \sum_k \big(f(\boldsymbol{x}_k) - y_k\big)^2 + c\,R(f)$ is motivated by the fact that we often have control over $\epsilon$ as a user-specified stopping criterion. In fact, in the era of overparameterized models, we can often fit training data to a desired $\epsilon$-optimal loss value [38]. However, if we adopt the unconstrained setting, it is unclear what value of $c$ would correspond to a particular stopping criterion.

[2]This assumption simplifies the exposition. If the null space is non-empty, one can still utilize it using [10].

[3]We assume that the Green's function exists and is continuous. Detailed treatment of existence conditions is beyond the scope of this work and can be found in text books such as [7].

[4]This property of $\boldsymbol{G}$ comes from the fact that $u$ is a positive definite kernel (definite instead of semi-definite, due to empty null space assumption on the operator $L$), thus so is its inverse (i.e. $g$). Since $g$ is a kernel, its associated Gram matrix is positive definite.

[5]$c_t > 0$ following from the assumption $\|\boldsymbol{z}_t\| > \sqrt{K}\epsilon$ and (7).

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
