[Supplementary Material]

# Self-Distillation Amplifies Regularization in Hilbert Space
# Supplementary Appendix

Hossein Mobahi♣     Mehrdad Farajtabar§     Peter L. Bartlett♣‡

hmobahi@google.com    farajtabar@google.com    bartlett@eecs.berkeley.edu

♣ Google Research, Mountain View, CA, USA
§ DeepMind, Mountain View, CA, USA
‡ EECS Dept., University of California at Berkeley, Berkeley, CA, USA

# A  Solving the Variational Problem

In this section we derive the solution to the following variational problem,

$$f^* \triangleq \arg\min_{f \in \mathcal{F}} \frac{1}{K} \sum_k \Big(f(\boldsymbol{x}_k) - y_k\Big)^2 + c \int_{\mathcal{X}} \int_{\mathcal{X}} u(\boldsymbol{x}, \boldsymbol{x}^\dagger) f(\boldsymbol{x}) f(\boldsymbol{x}^\dagger) \, d\boldsymbol{x} \, d\boldsymbol{x}^\dagger . \tag{26}$$

Using Dirac delta function, we can rewrite the objective function as,

$$f^* = \arg\min_{f \in \mathcal{F}} \frac{1}{K} \sum_k \Big(\int_{\mathcal{X}} f(\boldsymbol{x})\delta(\boldsymbol{x} - \boldsymbol{x}_k) \, d\boldsymbol{x} - y_k\Big)^2 + c \int_{\mathcal{X}} \int_{\mathcal{X}} u(\boldsymbol{x}, \boldsymbol{x}^\dagger) f(\boldsymbol{x}) f(\boldsymbol{x}^\dagger) \, d\boldsymbol{x} \, d\boldsymbol{x}^\dagger . \tag{27}$$

For brevity, name the objective functional $J$,

$$J(f) \triangleq \frac{1}{K} \sum_k \Big(\int_{\mathcal{X}} f(\boldsymbol{x})\delta(\boldsymbol{x} - \boldsymbol{x}_k) \, d\boldsymbol{x} - y_k\Big)^2 + c \int_{\mathcal{X}} \int_{\mathcal{X}} u(\boldsymbol{x}, \boldsymbol{x}^\dagger) f(\boldsymbol{x}) f(\boldsymbol{x}^\dagger) \, d\boldsymbol{x} \, d\boldsymbol{x}^\dagger . \tag{28}$$

If $f^*$ minimizes the $J(f)$, it must be a stationary point of $J$. That is, $J(f + \epsilon\phi) = J(f)$, for any $\phi \in \mathcal{F}$ as $\epsilon \to 0$. More precisely, it is necessary for $f^*$ to satisfy,

$$\forall \phi \in \mathcal{F}; \; \Big(\frac{d}{d\epsilon} J(f^* + \epsilon\phi)\Big)_{\epsilon=0} = 0 . \tag{29}$$

We first construct $J(f^* + \epsilon\phi)$,

$$J(f^* + \epsilon\phi) \;=\; \frac{1}{K} \sum_k \Big(\int_{\mathcal{X}} [f^* + \epsilon\phi](\boldsymbol{x})\delta(\boldsymbol{x} - \boldsymbol{x}_k) \, d\boldsymbol{x} - y_k\Big)^2 \tag{30}$$

$$+ \;\; c \int_{\mathcal{X}} \int_{\mathcal{X}} u(\boldsymbol{x}, \boldsymbol{x}^\dagger)[f^* + \epsilon\phi](\boldsymbol{x})[f^* + \epsilon\phi](\boldsymbol{x}^\dagger) \, d\boldsymbol{x} \, d\boldsymbol{x}^\dagger , \tag{31}$$

or equivalently,

$$J(f^* + \epsilon\phi) \;=\; \frac{1}{K} \sum_k \Big(\int_{\mathcal{X}} \big(f^*(\boldsymbol{x}) + \epsilon\phi(\boldsymbol{x})\big)\delta(\boldsymbol{x} - \boldsymbol{x}_k) \, d\boldsymbol{x} - y_k\Big)^2 \tag{32}$$

$$+ \;\; c \int_{\mathcal{X}} \int_{\mathcal{X}} u(\boldsymbol{x}, \boldsymbol{x}^\dagger)\big(f^*(\boldsymbol{x}) + \epsilon\phi(\boldsymbol{x})\big) \big(f^*(\boldsymbol{x}^\dagger) + \epsilon\phi(\boldsymbol{x}^\dagger)\big) \, d\boldsymbol{x}^\dagger \Big) d\boldsymbol{x} \, d\boldsymbol{x}^\dagger . \tag{33}$$

Thus,

$$\frac{d}{d\epsilon} J(f^* + \epsilon\phi) \tag{34}$$

$$= \;\; \frac{1}{K} \sum_k 2\Big(\int_{\mathcal{X}} \big(f^*(\boldsymbol{x}^\diamond) + \epsilon\phi(\boldsymbol{x}^\diamond)\big)\delta(\boldsymbol{x}^\diamond - \boldsymbol{x}_k) \, d\boldsymbol{x}^\diamond - y_k\Big)\Big(\int_{\mathcal{X}} \phi(\boldsymbol{x})\delta(\boldsymbol{x} - \boldsymbol{x}_k) \, d\boldsymbol{x}\Big) \tag{35}$$

$$+ \;\; c \int_{\mathcal{X}} \int_{\mathcal{X}} u(\boldsymbol{x}, \boldsymbol{x}^\dagger)\Big( \phi(\boldsymbol{x}) \big(f^*(\boldsymbol{x}^\dagger) + \epsilon\phi(\boldsymbol{x}^\dagger)\big) + \phi(\boldsymbol{x}^\dagger) \big(f^*(\boldsymbol{x}) + \epsilon\phi(\boldsymbol{x})\big) \Big) d\boldsymbol{x} \, d\boldsymbol{x}^\dagger . \tag{36}$$

Setting $\epsilon = 0$,

$$\Big(\frac{d}{d\epsilon} J(f^* + \epsilon\phi)\Big)_{\epsilon=0} \;=\; \frac{1}{K} \sum_k 2\Big(\int_{\mathcal{X}} f^*(\boldsymbol{x}^\diamond)\delta(\boldsymbol{x}^\diamond - \boldsymbol{x}_k) \, d\boldsymbol{x}^\diamond - y_k\Big)\Big(\int_{\mathcal{X}} \phi(\boldsymbol{x})\delta(\boldsymbol{x} - \boldsymbol{x}_k) \, d\boldsymbol{x}\Big) \tag{37}$$

$$+ \;\; c \int_{\mathcal{X}} \int_{\mathcal{X}} u(\boldsymbol{x}, \boldsymbol{x}^\dagger)\Big( \phi(\boldsymbol{x}) f^*(\boldsymbol{x}^\dagger) + \phi(\boldsymbol{x}^\dagger) f^*(\boldsymbol{x}) \Big) d\boldsymbol{x} \, d\boldsymbol{x}^\dagger . \tag{38}$$

By the symmetry of $u$,

$$\Big(\frac{d}{d\epsilon} J(f^* + \epsilon\phi)\Big)_{\epsilon=0} \;=\; \frac{1}{K} \sum_k 2\Big(\int_{\mathcal{X}} f^*(\boldsymbol{x}^\diamond)\delta(\boldsymbol{x}^\diamond - \boldsymbol{x}_k) \, d\boldsymbol{x}^\diamond - y_k\Big)\Big(\int_{\mathcal{X}} \phi(\boldsymbol{x})\delta(\boldsymbol{x} - \boldsymbol{x}_k) \, d\boldsymbol{x}\Big) \tag{39}$$

$$+ \;\; 2c \int_{\mathcal{X}} \int_{\mathcal{X}} u(\boldsymbol{x}, \boldsymbol{x}^\dagger) \, \phi(\boldsymbol{x}) \, f^*(\boldsymbol{x}^\dagger) \, d\boldsymbol{x} \, d\boldsymbol{x}^\dagger . \tag{40}$$

Factoring out $\phi$,

$$\Big(\frac{d}{d\epsilon} J(f^* + \epsilon\phi)\Big)_{\epsilon=0} = \int_{\mathcal{X}} 2\phi(\boldsymbol{x})\Big( \quad \frac{1}{K} \sum_k \delta(\boldsymbol{x} - \boldsymbol{x}_k) \Big(\int_{\mathcal{X}} f^*(\boldsymbol{x}^\diamond)\delta(\boldsymbol{x}^\diamond - \boldsymbol{x}_k) \, d\boldsymbol{x}^\diamond - y_k\Big) \tag{41}$$

$$+ \;\; c \int_{\mathcal{X}} u(\boldsymbol{x}, \boldsymbol{x}^\dagger) f^*(\boldsymbol{x}^\dagger) \, d\boldsymbol{x}^\dagger \quad \Big) d\boldsymbol{x} . \tag{42}$$

In order for the above to be zero for $\forall \phi \in \mathcal{F}$, it is necessary that,

$$\frac{1}{K} \sum_k \delta(\boldsymbol{x} - \boldsymbol{x}_k) \left( \int_\mathcal{X} f^*(\boldsymbol{x}^\diamond) \delta(\boldsymbol{x}^\diamond - \boldsymbol{x}_k) \, d\boldsymbol{x}^\diamond - y_k \right) + c \int_\mathcal{X} u(\boldsymbol{x}, \boldsymbol{x}^\dagger) f^*(\boldsymbol{x}^\dagger) \, d\boldsymbol{x}^\dagger = 0, \qquad (43)$$

which further simplifies to,

$$\frac{1}{K} \sum_k \delta(\boldsymbol{x} - \boldsymbol{x}_k) \left( f^*(\boldsymbol{x}_k) - y_k \right) + c \int_\mathcal{X} u(\boldsymbol{x}, \boldsymbol{x}^\dagger) f^*(\boldsymbol{x}^\dagger) \, d\boldsymbol{x}^\dagger = 0. \qquad (44)$$

We can equivalently express (44) by the following system of equations,

$$\begin{cases} \frac{1}{K} \sum_k \delta(\boldsymbol{x} - \boldsymbol{x}_k) r_k + c \int_\mathcal{X} u(\boldsymbol{x}, \boldsymbol{x}^\dagger) f^*(\boldsymbol{x}^\dagger) \, d\boldsymbol{x}^\dagger = 0 \\ r_1 = f^*(\boldsymbol{x}_1) - y_1 \\ \vdots \\ r_K = f^*(\boldsymbol{x}_K) - y_K \end{cases} \qquad (45)$$

We first focus on solving the first equation in $f^*$,

$$\frac{1}{K} \sum_k \delta(\boldsymbol{x} - \boldsymbol{x}_k) r_k + c \int_\mathcal{X} u(\boldsymbol{x}, \boldsymbol{x}^\dagger) f^*(\boldsymbol{x}^\dagger) \, d\boldsymbol{x}^\dagger = 0; \qquad (46)$$

later we can replace the resulted $f^*$ in other equations to obtain $r_k$'s. Let $g(\boldsymbol{x}, \boldsymbol{t})$ be a function such that,

$$\int_\mathcal{X} u(\boldsymbol{x}, \boldsymbol{x}^\dagger) g(\boldsymbol{x}^\dagger, \boldsymbol{t}) \, d\boldsymbol{x}^\dagger = \delta(\boldsymbol{x} - \boldsymbol{t}). \qquad (47)$$

Such $g$ is called the ***Green's function*** of the linear operator $L$ satisfying $[Lf](\boldsymbol{x}) = \int_\mathcal{X} u(\boldsymbol{x}, \boldsymbol{x}^\dagger) f(\boldsymbol{x}^\dagger) \, d\boldsymbol{x}^\dagger$. If we multiply both sides of (47) by $\frac{1}{K} \sum_k \delta(\boldsymbol{t} - \boldsymbol{x}_k) r_k$ and then integrate w.r.t. $\boldsymbol{t}$, we obtain,

$$\int_\mathcal{X} \left( \frac{1}{K} \sum_k r_k \delta(\boldsymbol{t} - \boldsymbol{x}_k) \int_\mathcal{X} u(\boldsymbol{x}, \boldsymbol{x}^\dagger) g(\boldsymbol{x}^\dagger, \boldsymbol{t}) \, d\boldsymbol{x}^\dagger \right) d\boldsymbol{t} \qquad (48)$$

$$= \int_\mathcal{X} \left( \frac{1}{K} \sum_k r_k \delta(\boldsymbol{t} - \boldsymbol{x}_k) \delta(\boldsymbol{x} - \boldsymbol{t}) \right) d\boldsymbol{t}. \qquad (49)$$

Rearranging the left hand side leads to,

$$\int_\mathcal{X} u(\boldsymbol{x}, \boldsymbol{x}^\dagger) \left( \frac{1}{K} \sum_k \int_\mathcal{X} r_k \delta(\boldsymbol{t} - \boldsymbol{x}_k) g(\boldsymbol{x}^\dagger, \boldsymbol{t}) \, d\boldsymbol{t} \right) d\boldsymbol{x}^\dagger \qquad (50)$$

$$= \int_\mathcal{X} \left( \frac{1}{K} \sum_k r_k \delta(\boldsymbol{t} - \boldsymbol{x}_k) \delta(\boldsymbol{x} - \boldsymbol{t}) \right) d\boldsymbol{t}. \qquad (51)$$

Using the sifting property of the delta function this simplifies to,

$$\int_\mathcal{X} u(\boldsymbol{x}, \boldsymbol{x}^\dagger) \left( \frac{1}{K} \sum_k r_k g(\boldsymbol{x}^\dagger, \boldsymbol{x}_k) \right) d\boldsymbol{x}^\dagger = \frac{1}{K} \sum_k r_k \delta(\boldsymbol{x} - \boldsymbol{x}_k). \qquad (52)$$

We can now use the above identity to eliminate $\frac{1}{K} \sum_k r_k \delta(\boldsymbol{x} - \boldsymbol{x}_k)$ in (46) and thus obtain,

$$\int_\mathcal{X} u(\boldsymbol{x}, \boldsymbol{x}^\dagger) \left( \frac{1}{K} \sum_k r_k g(\boldsymbol{x}^\dagger, \boldsymbol{x}_k) \right) d\boldsymbol{x}^\dagger + c \int_\mathcal{X} u(\boldsymbol{x}, \boldsymbol{x}^\dagger) f^*(\boldsymbol{x}^\dagger) \, d\boldsymbol{x}^\dagger = 0, \qquad (53)$$

or equivalently

$$\int_\mathcal{X} u(\boldsymbol{x}, \boldsymbol{x}^\dagger) \left( \frac{1}{K} \sum_k r_k g(\boldsymbol{x}^\dagger, \boldsymbol{x}_k) + c \, f^*(\boldsymbol{x}^\dagger) \right) d\boldsymbol{x}^\dagger = 0. \qquad (54)$$

A sufficient (and also necessary, as $u$ is assumed to have empty null space) for the above to hold is that,

$$f^*(\boldsymbol{x}) = -\frac{1}{cK} \sum_k r_k g(\boldsymbol{x}, \boldsymbol{x}_k). \qquad (55)$$

We can now eliminate $f^*$ in the system of equations (45) and obtain a system that only depends on $r_k$'s,

$$\begin{cases} r_1 = -\frac{1}{cK} \sum_k r_k g(\boldsymbol{x}_1, \boldsymbol{x}_k) - y_1 \\ \vdots \\ r_K = -\frac{1}{cK} \sum_k r_k g(\boldsymbol{x}_K, \boldsymbol{x}_k) - y_K \end{cases} \qquad (56)$$

This is a linear system in $r_k$ and can be expressed in vector/matrix form,

$$(c\boldsymbol{I} + \boldsymbol{G})\boldsymbol{r} = -c\,\boldsymbol{y}\,. \tag{57}$$

Thus,

$$\boldsymbol{r} = -c\,(c\boldsymbol{I} + \boldsymbol{G})^{-1}\boldsymbol{y}\,, \tag{58}$$

and finally using the definition of $f^*$ in (55) we obtain,

$$f^*(\boldsymbol{x}) \;=\; -\frac{1}{c}\,\boldsymbol{g}_{\boldsymbol{x}}^T\,\boldsymbol{r} \;=\; \boldsymbol{g}_{\boldsymbol{x}}^T\,(c\boldsymbol{I} + \boldsymbol{G})^{-1}\boldsymbol{y}\,. \tag{59}$$

# B    Equivalent Kernel Regression Problem

Given a positive definite kernel function $g(\,.\,,\,.\,)$. Recall that the solution of regularized kernel regression after $t$ rounds of self-distillation has the form,

$$f_t^*(\boldsymbol{x}) = \boldsymbol{g}_{\boldsymbol{x}}^T \boldsymbol{G}^t \Pi_{i=0}^t (\boldsymbol{G} + c_i \boldsymbol{I})^{-1} \boldsymbol{y}_0 \,. \tag{60}$$

On the other hand, the solution to a standard kernel ridge regression on the same training data with a positive definite kernel $g^\dagger$ has the form,

$$f^\dagger(\boldsymbol{x}) = \boldsymbol{g}_{\boldsymbol{x}}^{\dagger \, T} (\boldsymbol{G}^\dagger + c_0 \boldsymbol{I})^{-1} \boldsymbol{y}_0 \,, \tag{61}$$

for which there are standard generalization bounds. We claim $f_t^*$ can be equivalently written in this standard form by a proper choice of $g^\dagger$ (as a function of $g$). As a result of that, we show the spectrum of the Gram matrix $\boldsymbol{G}^\dagger$ relates to that of $\boldsymbol{G}$ via,

$$\lambda_k^\dagger = c_0 \frac{1}{\frac{\Pi_{i=0}^t (\lambda_k + c_i)}{\lambda_k^{t+1}} - 1} \,. \tag{62}$$

Our strategy for tackling this problem is inspired by the proof technique in Corollary 6.7 of [3]. Let $P$ be the data-dependent linear operator defined as,

$$[Ph](\boldsymbol{x}) \triangleq \frac{1}{K} \sum_{k=1}^K h(\boldsymbol{x}_k) g(\boldsymbol{x}, \boldsymbol{x}_k) \,. \tag{63}$$

Let $\mathcal{H}$ denote the Reproducing Kernel Hilbert Space associated with $g$ and $\langle\,.\,,\,.\,\rangle_{\mathcal{H}}$ be the dot product in $\mathcal{H}$. It is easy to verify that $P$ is a ***positive definite operator*** in this space, i.e. it satisfies $\langle h\,,\, Ph \rangle > 0$ for any $h \in \mathcal{H}$ due to,

$$\langle h\,,\, Ph \rangle_{\mathcal{H}} \;=\; \langle h\,,\, \frac{1}{K} \sum_{k=1}^K h(\boldsymbol{x}_k) g(.,\boldsymbol{x}_k) \rangle \tag{64}$$

$$=\; \frac{1}{K} \sum_{k=1}^K h(\boldsymbol{x}_k) \underbrace{\langle h\,,\, g(.,\boldsymbol{x}_k) \rangle}_{h(\boldsymbol{x}_k)} \tag{65}$$

$$=\; \frac{1}{K} \sum_{k=1}^K h^2(\boldsymbol{x}_k) > 0 \,, \tag{66}$$

where we used $\langle h\,,\, g(.,\boldsymbol{x}) \rangle = h(\boldsymbol{x})$ due to the reproducing property of $\mathcal{H}$. Since $P$ is positive definite, there exist eigenfunctions $\phi_j$ and eigenvalues $\lambda_j \geq 0$ that satisfy $[P\phi_j](\boldsymbol{x}) = \lambda_j \phi_j(\boldsymbol{x})$. Plugging the definition of $P$ into this identity yields,

$$\frac{1}{K} \sum_{k=1}^K \phi_j(\boldsymbol{x}_k) g(\boldsymbol{x}, \boldsymbol{x}_k) = \lambda_j \phi_j(\boldsymbol{x}) \,. \tag{67}$$

In particular, evaluating the latter identity at the points $\boldsymbol{x} \in \cup_{p=1}^K \{\boldsymbol{x}_p\}$ gives $\frac{1}{K} \sum_{k=1}^K \phi_j(\boldsymbol{x}_k) g(\boldsymbol{x}_p, \boldsymbol{x}_k) = \lambda_j \phi_j(\boldsymbol{x}_p)$ for $p = 1, \dots, K$. Recalling that $\boldsymbol{G}$ is evaluation of $\frac{1}{K} g(\,.\,,\,.\,)$ at pairs of points across $\cup_{k=1}^K \{\boldsymbol{x}_k\}$, this identity be expressed equivalently as,

$$\boldsymbol{G}\phi_j = \lambda_j \phi_j \,. \tag{68}$$

This implies $\phi_j$ is an eigenvector of $\boldsymbol{G}$ with corresponding eigenvalue of $\lambda_j$ for any $j$ that $\boldsymbol{G}\phi_j \neq 0$. Thus, by sorting $\phi_j$ in non-increasing order of $\lambda_j$, and placing them for $j = 1, \dots, K$ into the matrix $\boldsymbol{\Phi}$ and the diagonal matrix $\boldsymbol{\Lambda}$ respectively, we obtain,

$$\boldsymbol{\Phi} = \boldsymbol{V} \quad,\quad \boldsymbol{\Lambda} = \boldsymbol{D} \,. \tag{69}$$

Since the eigenvectors of $\boldsymbol{G}^t \Pi_{i=0}^t (\boldsymbol{G} + c_i \boldsymbol{I})^{-1}$ are the same as those of $\boldsymbol{G}$ (adding a multiple of $\boldsymbol{I}$ or applying matrix inversion do not change eigenvectors), and the eigenvectors of $\boldsymbol{G}$ as showed in (68) are $\boldsymbol{\Phi}$, we can write,

$$\boldsymbol{G}^t \Pi_{i=0}^t (\boldsymbol{G} + c_i \boldsymbol{I})^{-1} = \boldsymbol{\Phi}^T \boldsymbol{\Lambda}^t \Pi_{i=0}^t (\boldsymbol{\Lambda} + c_i \boldsymbol{I})^{-1} \boldsymbol{\Phi} \,. \tag{70}$$

On the other hand, using the same vector notation and recalling that $\boldsymbol{g}$ is the evaluation of $\frac{1}{K} g(\,.\,,\,\boldsymbol{x}_k)$ at $k = 1, \dots, K$, we can express (67) as $\phi_j^T \boldsymbol{g}_{\boldsymbol{x}} = \lambda_j \phi_j(\boldsymbol{x})$. Expressing this simultaneously for $j = 1, \dots, K$ yields $\boldsymbol{\Phi}\boldsymbol{g}_{\boldsymbol{x}} = \boldsymbol{\Lambda}\phi_{\boldsymbol{x}}$, or equivalently

$$\boldsymbol{g}_{\boldsymbol{x}} = \boldsymbol{\Phi}^T \boldsymbol{\Lambda} \phi_{\boldsymbol{x}} \,, \tag{71}$$

where $\phi_{\boldsymbol{x}} \triangleq [\phi_1(\boldsymbol{x}), \dots, \phi_K(\boldsymbol{x})]$. Plugging (70) and (71) with into (60) gives,

$$f_t^*(\boldsymbol{x}) \;=\; \boldsymbol{g}_{\boldsymbol{x}}^T \boldsymbol{G}^t \Pi_{i=0}^t (\boldsymbol{G} + c_i \boldsymbol{I})^{-1} \boldsymbol{y}_0 \tag{72}$$

$$=\; \phi_{\boldsymbol{x}}^T \boldsymbol{\Lambda} \boldsymbol{\Phi} \boldsymbol{\Phi}^T \boldsymbol{\Lambda}^t \Pi_{i=0}^t (\boldsymbol{\Lambda} + c_i \boldsymbol{I})^{-1} \boldsymbol{\Phi} \boldsymbol{y}_0 \tag{73}$$

$$=\; \phi_{\boldsymbol{x}}^T \boldsymbol{\Lambda}^{t+1} \Pi_{i=0}^t (\boldsymbol{\Lambda} + c_i \boldsymbol{I})^{-1} \boldsymbol{\Phi} \boldsymbol{y}_0 \,. \tag{74}$$

Suppose $g^\dagger$ is a positive definite kernel and let $[P^\dagger h](x) \triangleq \frac{1}{K} \sum_{k=1}^{K} h(\boldsymbol{x}_k) g^\dagger(\boldsymbol{x}, \boldsymbol{x}_i)$. We assume the operator $P^\dagger$ shares the same eigenfunction as those of $P$, but varies in its eigenvalues $\lambda_j^\dagger \geq 0$, i.e. $[P^\dagger \phi_j](\boldsymbol{x}) = \lambda_j^\dagger \phi_j(\boldsymbol{x})$. Thus, by a similar argument, the solution of (61) can be written as,

$$f^\dagger(\boldsymbol{x}) = \boldsymbol{\phi}_{\boldsymbol{x}}^T \boldsymbol{\Lambda}^\dagger (\boldsymbol{\Lambda}^\dagger + c_0 \boldsymbol{I})^{-1} \boldsymbol{\Phi} \boldsymbol{y}_0 \,, \tag{75}$$

Thus in order to have $f^\dagger = f_t^*$, it is sufficient to have,

$$\boldsymbol{\Lambda}^{t+1} \Pi_{i=0}^t (\boldsymbol{\Lambda} + c_i \boldsymbol{I})^{-1} = \boldsymbol{\Lambda}^\dagger (\boldsymbol{\Lambda}^\dagger + c_0 \boldsymbol{I})^{-1} \,. \tag{76}$$

Since the matrices above are all diagonal, this can be expressed equivalently as,

$$\frac{\lambda_k^{t+1}}{\Pi_{i=0}^t (\lambda_k + c_i)} = \frac{\lambda_k^\dagger}{\lambda_k^\dagger + c_0} \,. \tag{77}$$

Solving in $\lambda_k^\dagger$ yields,

$$\lambda_k^\dagger = c_0 \frac{1}{\frac{\Pi_{i=0}^t (\lambda_k + c_i)}{\lambda_k^{t+1}} - 1} \,. \tag{78}$$

Note that this is a valid solution for $\lambda_k^\dagger$, i.e. it satisfies the requirement $\lambda_k^\dagger \geq 0$. This is because $\omega_k \triangleq \frac{\lambda_k^{t+1}}{\Pi_{i=0}^t (\lambda_k + c_i)}$ always satisfies[6] $0 < \omega_k < 1$ and that the function $\lambda_k^\dagger(\omega_k) \triangleq c_0 \frac{1}{\frac{1}{\omega_k} - 1}$ is well-defined ($\omega_k \neq 0$) and is increasing when $0 < \omega_k < 1$.

# C  Proofs

**Proposition 1** *The variational problem (9) has a solution of the form,*

$$f^*(\boldsymbol{x}) = \boldsymbol{g}_{\boldsymbol{x}}^T(c\boldsymbol{I} + \boldsymbol{G})^{-1}\boldsymbol{y}. \tag{79}$$

See Appendix A for a proof.

**Proposition 2** *The following identity holds,*

$$\frac{1}{K}\sum_k \left(f^*(\boldsymbol{x}_k) - y_k\right)^2 = \frac{1}{K}\sum_k (z_k\frac{c}{c+d_k})^2. \tag{80}$$

**Proof**

$$\frac{1}{K}\left(f^*(\boldsymbol{x}_k) - y_k\right)^2 \tag{81}$$

$$= \frac{1}{K}\left(\boldsymbol{g}_{\boldsymbol{x}_k}^T(c\boldsymbol{I}+\boldsymbol{G})^{-1}\boldsymbol{y} - y_k\right)^2 \tag{82}$$

$$= \frac{1}{K}\left\|\boldsymbol{G}(c\boldsymbol{I}+\boldsymbol{G})^{-1}\boldsymbol{y} - \boldsymbol{y}\right\|^2 \tag{83}$$

$$= \frac{1}{K}\left\|\boldsymbol{V}^T\boldsymbol{D}(c\boldsymbol{I}+\boldsymbol{D})^{-1}\boldsymbol{V}\boldsymbol{y} - \boldsymbol{y}\right\|^2, \tag{84}$$

which after exploiting rotation invariance property of $\|.\|$ and the fact that the matrix of eigenvectors $\boldsymbol{V}$ is a rotation matrix, can be expressed as,

$$\frac{1}{K}\left(f^*(\boldsymbol{x}_k) - y_k\right)^2 \tag{85}$$

$$= \frac{1}{K}\left\|\boldsymbol{V}^T\boldsymbol{D}(c\boldsymbol{I}+\boldsymbol{D})^{-1}\boldsymbol{V}\boldsymbol{y} - \boldsymbol{y}\right\|^2 \tag{86}$$

$$= \frac{1}{K}\left\|\boldsymbol{V}\boldsymbol{V}^T\boldsymbol{D}(c\boldsymbol{I}+\boldsymbol{D})^{-1}\boldsymbol{V}\boldsymbol{y} - \boldsymbol{V}\boldsymbol{y}\right\|^2 \tag{87}$$

$$= \frac{1}{K}\left\|\boldsymbol{D}(c\boldsymbol{I}+\boldsymbol{D})^{-1}\boldsymbol{z} - \boldsymbol{z}\right\|^2 \tag{88}$$

$$= \frac{1}{K}\left\|\left(\boldsymbol{D}(c\boldsymbol{I}+\boldsymbol{D})^{-1} - \boldsymbol{I}\right)\boldsymbol{z}\right\|^2 \tag{89}$$

$$= \frac{1}{K}\sum_k(\frac{d_k}{c+d_k} - 1)^2 z_k^2 \tag{90}$$

$$= \frac{1}{K}\sum_k(z_k\frac{c}{c+d_k})^2, \tag{91}$$

$\square$

**Proposition 3** *For any $t \geq 0$, if $\|\boldsymbol{z}_i\| > \sqrt{K}\,\epsilon$ for $i = 0, \ldots, t$, then,*

$$\|\boldsymbol{z}_t\| \geq a^t(\kappa)\|\boldsymbol{z}_0\| - \sqrt{K}\,\epsilon\,b(\kappa)\frac{a^t(\kappa) - 1}{a(\kappa) - 1}, \tag{92}$$

*where,*

$$a(x) \triangleq \frac{(r_0 - 1)^2 + x(2r_0 - 1)}{(r_0 - 1 + x)^2} \tag{93}$$

$$b(x) \triangleq \frac{r_0^2 x}{(r_0 - 1 + x)^2} \tag{94}$$

$$r_0 \triangleq \frac{1}{\sqrt{K}\,\epsilon}\|\boldsymbol{z}_0\| \quad, \quad \kappa \triangleq \frac{d_{\max}}{d_{\min}}. \tag{95}$$

**Proof** We start from the identity we obtained in (17). By diving both sides of it by $\sqrt{K}\,\epsilon$ we obtain,

$$\frac{1}{\sqrt{K}\,\epsilon}\boldsymbol{z}_t = \boldsymbol{D}(\frac{\alpha_t\sqrt{K}\,\epsilon}{\|\boldsymbol{z}_{t-1}\| - \sqrt{K}\,\epsilon}\boldsymbol{I} + \boldsymbol{D})^{-1}\frac{1}{\sqrt{K}\,\epsilon}\boldsymbol{z}_{t-1}, \tag{96}$$

where,

$$d_{\min} \leq \alpha_t \leq d_{\max}. \tag{97}$$

Note that the matrix $\boldsymbol{D}(\frac{\alpha_t\sqrt{K\epsilon}}{\|\boldsymbol{z}_{t-1}\|-\sqrt{K\epsilon}}\boldsymbol{I}+\boldsymbol{D})^{-1}$ in the above identitiy is *diagonal* and its $k$'th entry can be expressed as,

$$\left(\boldsymbol{D}(\frac{\alpha_t\sqrt{K\epsilon}}{\|\boldsymbol{z}_{t-1}\|-\sqrt{K\epsilon}}\boldsymbol{I}+\boldsymbol{D})^{-1}\right)[k,k] = \frac{d_k}{\frac{\alpha_t\sqrt{K\epsilon}}{\|\boldsymbol{z}_{t-1}\|-\sqrt{K\epsilon}}+d_k} = \frac{1}{\frac{\frac{\alpha_t}{d_k}}{\frac{\|\boldsymbol{z}_{t-1}\|}{\sqrt{K\epsilon}}-1}+1}. \tag{98}$$

Thus, as long as $\|\boldsymbol{z}_{t-1}\| > \sqrt{K\epsilon}$ we can get the following upper and lower bounds,

$$\frac{1}{\frac{\frac{d_{\max}}{d_{\min}}}{\frac{\|\boldsymbol{z}_{t-1}\|}{\sqrt{K\epsilon}}-1}+1} \leq \left(\boldsymbol{D}(\frac{\alpha_t\sqrt{K\epsilon}}{\|\boldsymbol{z}_{t-1}\|-\sqrt{K\epsilon}}\boldsymbol{I}+\boldsymbol{D})^{-1}\right)[k,k] \leq \frac{1}{\frac{\frac{d_{\min}}{d_{\max}}}{\frac{\|\boldsymbol{z}_{t-1}\|}{\sqrt{K\epsilon}}-1}+1}. \tag{99}$$

Putting the above fact beside recurrence relation of $\boldsymbol{z}_t$ in (96), we can bound $\frac{1}{\sqrt{K\epsilon}}\|\boldsymbol{z}_t\|$ as,

$$\frac{1}{\frac{\kappa}{r_{t-1}-1}+1}r_{t-1} \leq r_t \leq \frac{1}{\frac{\frac{1}{\kappa}}{r_{t-1}-1}+1}r_{t-1}, \tag{100}$$

where we used short hand notation,

$$\kappa \triangleq \frac{d_{\max}}{d_{\min}} \tag{101}$$

$$r_t \triangleq \frac{1}{\sqrt{K\epsilon}}\|\boldsymbol{z}_t\|. \tag{102}$$

Note that $\kappa$ is the ***condition number*** of the matrix $\boldsymbol{G}$ and by definition satisfies $\kappa \geq 1$. To further simplify the bounds, we use the inequality[7],

$$\frac{1}{\frac{\frac{1}{\kappa}}{r_{t-1}-1}+1}r_{t-1} \leq r_{t-1}\frac{(r_0-1)^2+\frac{1}{\kappa}(2r_0-1)}{(r_0-1+\frac{1}{\kappa})^2} - \frac{r_0^2\frac{1}{\kappa}}{(r_0-1+\frac{1}{\kappa})^2}, \tag{103}$$

and[8],

$$\frac{1}{\frac{\kappa}{r_{t-1}-1}+1}r_{t-1} \geq r_{t-1}\frac{(r_0-1)^2+\kappa(2r_0-1)}{(r_0-1+\kappa)^2} - \frac{r_0^2\kappa}{(r_0-1+\kappa)^2}. \tag{104}$$

For brevity, we introduce,

$$a(x) \triangleq \frac{(r_0-1)^2+x(2r_0-1)}{(r_0-1+x)^2} \tag{105}$$

$$b(x) \triangleq \frac{r_0^2x}{(r_0-1+x)^2}. \tag{106}$$

Therefore, the bounds can be expressed more concisely as,

$$a(\kappa)\,r_{t-1}-b(\kappa) \quad \leq \quad r_t \quad \leq \quad a(\frac{1}{\kappa})\,r_{t-1}-b(\frac{1}{\kappa}). \tag{107}$$

Now since both $r_{t-1} \triangleq \frac{1}{\sqrt{K\epsilon}}\|\boldsymbol{z}_{t-1}\|$ and $a(\kappa)$ or $a(\frac{1}{\kappa})$ are non-negative, we can solve the recurrence[9] and obtain,

$$a^t(\kappa)r_0-b(\kappa)\frac{a^t(\kappa)-1}{a(\kappa)-1} \quad \leq \quad r_t \quad \leq \quad a^t(\frac{1}{\kappa})r_0-b(\frac{1}{\kappa})\frac{a^t(\frac{1}{\kappa})-1}{a(\frac{1}{\kappa})-1}. \tag{108}$$

$\square$

**Proposition 4** *Starting from* $\|\boldsymbol{y}_0\| > \sqrt{K\,\epsilon}$*, meaningful (non-collapsing solution) self-distillation is possible at least for* $\underline{t}$ *rounds,*

$$\underline{t} \triangleq \frac{\frac{\|\boldsymbol{y}_0\|}{\sqrt{K\,\epsilon}} - 1}{\kappa} \,. \tag{109}$$

**Proof** Recall that the assumption $\|\boldsymbol{z}_t\| > \sqrt{K\,\epsilon}$ translates into $r_t > 1$. We now obtain a sufficient condition for $r_t > 1$ by requiring a lower bound on $r_t$ to be greater than one. For that purpose, we utilize the lower bound we established in (108),

$$\underline{r_t} \triangleq a^t(\kappa) r_0 - b(\kappa) \frac{a^t(\kappa) - 1}{a(\kappa) - 1} \,. \tag{110}$$

Setting the above to value 1 implies,

$$\underline{r_t} = 1 \quad \Rightarrow \quad t = \frac{\log\left(\frac{1 - a(\kappa) + b(\kappa)}{b(\kappa) + r_0(1 - a(\kappa))}\right)}{\log\left(a(\kappa)\right)} = \frac{\log\left(\frac{1 + \frac{\kappa - 1}{r_0^2}}{1 + \frac{\kappa - 1}{r_0}}\right)}{\log\left(1 - \frac{(\frac{\kappa - 1}{r_0} + \frac{1}{r_0})(\frac{\kappa - 1}{r_0})}{(1 + \frac{\kappa - 1}{r_0})^2}\right)} \,. \tag{111}$$

Observe that,

$$\frac{\log\left(\frac{1 + \frac{\kappa - 1}{r_0^2}}{1 + \frac{\kappa - 1}{r_0}}\right)}{\log\left(1 - \frac{(\frac{\kappa - 1}{r_0} + \frac{1}{r_0})(\frac{\kappa - 1}{r_0})}{(1 + \frac{\kappa - 1}{r_0})^2}\right)} \geq \frac{r_0 - 1}{\kappa} \,, \tag{112}$$

Thus,

$$t \geq \frac{r_0 - 1}{\kappa} = \frac{\frac{\|\boldsymbol{z}_0\|}{\sqrt{K\,\epsilon}} - 1}{\kappa} = \frac{\frac{\|\boldsymbol{z}_0\|}{\sqrt{K\,\epsilon}} - 1}{\kappa} = \frac{\frac{\|\boldsymbol{y}_0\|}{\sqrt{K\,\epsilon}} - 1}{\kappa} \,. \tag{113}$$

$\square$

**Theorem 5** *Suppose* $\|\boldsymbol{y}_0\| > \sqrt{K\,\epsilon}$ *and* $t \leq \frac{\|\boldsymbol{y}_0\|}{\kappa\sqrt{K\,\epsilon}} - \frac{1}{\kappa}$*. Then for any pair of diagonals of* $\boldsymbol{D}$*, namely* $d_j$ *and* $d_k$*, with the condition that* $d_k > d_j$*, the following inequality holds.*

$$\frac{\boldsymbol{B}_{t-1}[k,k]}{\boldsymbol{B}_{t-1}[j,j]} \geq \left(\frac{\frac{\|\boldsymbol{y}_0\|}{\sqrt{K\,\epsilon}} - 1 + \frac{d_{\min}}{d_j}}{\frac{\|\boldsymbol{y}_0\|}{\sqrt{K\,\epsilon}} - 1 + \frac{d_{\min}}{d_k}}\right)^t \,. \tag{114}$$

**Proof** We start with the definition of $\boldsymbol{A}_t$ from (13) and proceed as,

$$\frac{\boldsymbol{A}_t[k,k]}{\boldsymbol{A}_t[j,j]} = \frac{1 + \frac{c_t}{d_j}}{1 + \frac{c_t}{d_k}} \,. \tag{115}$$

Since the derivative of the r.h.s. above w.r.t. $c_t$ is non-negative as long as $d_k \geq d_j$, it is non-decreasing in $c_t$. Therefore, we can get a lower bound on r.h.s. using a lower bound on $c_t$ (denoted by $\underline{c_t}$),

$$\frac{\boldsymbol{A}_t[k,k]}{\boldsymbol{A}_t[j,j]} \geq \frac{1 + \frac{\underline{c_t}}{d_j}}{1 + \frac{\underline{c_t}}{d_k}} \,. \tag{116}$$

Also, since the assumption $t \leq \frac{\|\boldsymbol{y}_0\|}{\kappa\sqrt{K\,\epsilon}} - \frac{1}{\kappa}$ guarantees non-collapse conditions $c_t > 0$ and $\|\boldsymbol{z}_t\| > \sqrt{K\,\epsilon}$, we can apply (16) and have the following lower bound on $c_t$

$$c_t \geq \frac{d_{\min}\sqrt{K\,\epsilon}}{\|\boldsymbol{z}_t\| - \sqrt{K\,\epsilon}} \,. \tag{117}$$

Since the r.h.s. (117) is decreasing in $\|\boldsymbol{z}_t\|$, the smallest value for the r.h.s. is attained by the largest value of $\|\boldsymbol{z}_t\|$. However, as $\|\boldsymbol{z}_t\|$ is decreasing in $t$ (see beginning of Section 3.2), its largest value is attained at $t = 0$. Putting these together we obtain,

$$c_t \geq \frac{d_{\min}\sqrt{K\,\epsilon}}{\|\boldsymbol{z}_0\| - \sqrt{K\,\epsilon}} \,. \tag{118}$$

Using the r.h.s. of the above as $\underline{c_t}$ and applying it to (116) yields,

$$\frac{\boldsymbol{A}_t[k,k]}{\boldsymbol{A}_t[j,j]} \geq \frac{\frac{\|\boldsymbol{z}_0\|}{\sqrt{K\,\epsilon}} - 1 + \frac{d_{\min}}{d_j}}{\frac{\|\boldsymbol{z}_0\|}{\sqrt{K\,\epsilon}} - 1 + \frac{d_{\min}}{d_k}} \,. \tag{119}$$

Notice that both sides of the inequality are positive; $\boldsymbol{A}_t$ based on its definition in (13) and r.h.s. by the fact that $\|\boldsymbol{z}_0\| \geq \sqrt{K}\,\epsilon$. Therefore, we can instantiate the above inequality at each distillation step $i$, for $i = 0, \ldots, t-1$, and multiply them to obtain,

$$\Pi_{i=0}^{t-1} \frac{\boldsymbol{A}_i[k,k]}{\boldsymbol{A}_i[j,j]} \geq \Big( \frac{\frac{\|\boldsymbol{z}_0\|}{\sqrt{K}\,\epsilon} - 1 + \frac{d_{\min}}{d_j}}{\frac{\|\boldsymbol{z}_0\|}{\sqrt{K}\,\epsilon} - 1 + \frac{d_{\min}}{d_k}} \Big)^t . \tag{120}$$

or equivalently,

$$\frac{\boldsymbol{B}_{t-1}[k,k]}{\boldsymbol{B}_{t-1}[j,j]} \geq \Big( \frac{\frac{\|\boldsymbol{z}_0\|}{\sqrt{K}\,\epsilon} - 1 + \frac{d_{\min}}{d_j}}{\frac{\|\boldsymbol{z}_0\|}{\sqrt{K}\,\epsilon} - 1 + \frac{d_{\min}}{d_k}} \Big)^t . \tag{121}$$

$\square$

**Theorem 6** *Suppose $\|\boldsymbol{y}_0\| > \sqrt{K}\,\epsilon$. Then the sparsity index $S_{\boldsymbol{B}_{\underline{t}-1}}$ (where $\underline{t} = \frac{\|\boldsymbol{y}_0\|}{\kappa\sqrt{K}\,\epsilon} - \frac{1}{\kappa}$ is number of guaranteed self-distillation steps before solution collapse) "decreases" in $\epsilon$, i.e. lower $\epsilon$ yields higher sparsity.*

*Furthermore at the limit $\epsilon \to 0$, the sparsity index has the form,*

$$\lim_{\epsilon \to 0} S_{\boldsymbol{B}_{\underline{t}-1}} = e^{\frac{d_{\min}}{\kappa}\, \min_{k \in \{1,2,\ldots,K-1\}} \big(\frac{1}{d_k} - \frac{1}{d_{k+1}}\big)} . \tag{122}$$

**Proof** We first show that the sparsity index is decreasing in $\epsilon$. We start from the definition of the sparsity index $S_{\boldsymbol{B}_{\underline{t}-1}}$ in (24) which we repeat below,

$$S_{\boldsymbol{B}_{\underline{t}-1}} = \min_{k \in \{1,2,\ldots,K-1\}} \left( \frac{\frac{\|\boldsymbol{y}_0\|}{\sqrt{K}\,\epsilon} - 1 + \frac{d_{\min}}{d_k}}{\frac{\|\boldsymbol{y}_0\|}{\sqrt{K}\,\epsilon} - 1 + \frac{d_{\min}}{d_{k+1}}} \right)^{\frac{\|\boldsymbol{y}_0\|}{\kappa\sqrt{K}\,\epsilon} - \frac{1}{\kappa}} . \tag{123}$$

For brevity, we define base and exponent as,

$$b \triangleq \frac{m + \frac{d_{\min}}{d_k}}{m + \frac{d_{\min}}{d_{k+1}}} \tag{124}$$

$$p \triangleq \frac{m}{\kappa} \tag{125}$$

$$m \triangleq \frac{\|\boldsymbol{y}_0\|}{\sqrt{K}\,\epsilon} - 1 , \tag{126}$$

so that,

$$S_{\boldsymbol{B}_{\underline{t}-1}}(\epsilon) = b^p . \tag{127}$$

The derivative is thus,

$$\frac{d}{d\epsilon} S_{\boldsymbol{B}_{\underline{t}-1}} \tag{128}$$

$$= \frac{d\,S_{\boldsymbol{B}_{\underline{t}-1}}}{dm} \frac{dm}{d\epsilon} \tag{129}$$

$$= \Big( b^p \big( \frac{p\,b_m}{b} + p_m\,\log(b) \big) \Big) \Big( \frac{dm}{d\epsilon} \Big) \tag{130}$$

$$= b^p \big( \frac{p\,b_m}{b} + p_m\,\log(b) \big) \big( -\frac{1}{2\epsilon}(m+1) \big) \tag{131}$$

$$= b^p \big( \frac{p}{m + \frac{d_{\min}}{d_k}} - \frac{p}{m + \frac{d_{\min}}{a_{k+1}}} + \frac{1}{\kappa}\,\log(b) \big) \big( -\frac{1}{2\epsilon}(m+1) \big) \tag{132}$$

$$= \frac{b^p}{\kappa} \big( \frac{m}{m + \frac{d_{\min}}{d_k}} - \frac{m}{m + \frac{d_{\min}}{a_{k+1}}} + \log(b) \big) \big( -\frac{1}{2\epsilon}(m+1) \big) \tag{133}$$

$$= \frac{b^p}{\kappa} \big( \frac{1}{1 + \frac{d_{\min}}{m\,d_k}} - \frac{1}{1 + \frac{d_{\min}}{m\,a_{k+1}}} + \log(b) \big) \big( -\frac{1}{2\epsilon}(m+1) \big) \tag{134}$$

$$= \frac{b^p}{\kappa} \big( \frac{1}{1 + \frac{d_{\min}}{m\,d_k}} - \frac{1}{1 + \frac{d_{\min}}{m\,a_{k+1}}} + \log\big(\frac{1 + \frac{d_{\min}}{m\,d_k}}{1 + \frac{d_{\min}}{m\,d_{k+1}}}\big) \big) \big( -\frac{1}{2\epsilon}(m+1) \big) \tag{135}$$

$$= \frac{b^p}{\kappa} \big( \frac{1}{1 + \frac{d_{\min}}{m\,d_k}} + \log(1 + \frac{d_{\min}}{m\,d_k}) - \frac{1}{1 + \frac{d_{\min}}{m\,a_{k+1}}} - \log(1 + \frac{d_{\min}}{m\,d_{k+1}}) \big) \big( -\frac{1}{2\epsilon}(m+1) \big) . \tag{136}$$

We now focus on the first parentheses. Define the function $e(x) \triangleq \frac{1}{x} + \log(x)$. Thus we can write the contents in the first parentheses more compactly,

$$\frac{1}{1 + \frac{d_{\min}}{m\, d_k}} + \log(1 + \frac{d_{\min}}{m\, d_k}) - \frac{1}{1 + \frac{d_{\min}}{m\, a_{k+1}}} - \log(1 + \frac{d_{\min}}{m\, d_{k+1}}) \tag{137}$$

$$= \quad e(1 + \frac{d_{\min}}{m\, d_k}) - e(1 + \frac{d_{\min}}{m\, d_{k+1}}). \tag{138}$$

However, $e'(x) = \frac{x-1}{x^2}$, thus when $x > 1$ the function $e'(x)$ is positive. Consequently, when $x > 1$ $e(x)$ is increasing. In fact, since both $\frac{d_{\min}}{m\, d_k}$ and $\frac{d_{\min}}{m\, d_k}$ are positive, the arguments of $e$ satsify the condition of being greater that 1 and thus $e$ is increasing. On the other hand, since $d_{k+1} > d_k$ it follows that $1 + \frac{d_{\min}}{m\, d_k} > 1 + \frac{d_{\min}}{m\, d_{k+1}}$, and thus by leveraging the fact that $e$ is increasing we obtain $e(1 + \frac{d_{\min}}{m\, d_k}) > e(1 + \frac{d_{\min}}{m\, d_{k+1}})$. Finally by plugging the definition of $e$ we obtain,

$$\frac{1}{1 + \frac{d_{\min}}{m\, d_k}} + \log(1 + \frac{d_{\min}}{m\, d_k}) > \frac{1}{1 + \frac{d_{\min}}{m\, a_{k+1}}} + \log(1 + \frac{d_{\min}}{m\, d_{k+1}}). \tag{139}$$

It is now easy to determine the sign of $\frac{d}{d\epsilon} S$ as shown below,

$$\frac{d}{d\epsilon} S_{\boldsymbol{B}_{\underline{t}-1}} \tag{140}$$

$$= \quad \underbrace{\frac{b^p}{\kappa}}_{\text{positive}} \underbrace{\left( \frac{1}{1 + \frac{d_{\min}}{m\, d_k}} + \log(1 + \frac{d_{\min}}{m\, d_k}) - \frac{1}{1 + \frac{d_{\min}}{m\, a_{k+1}}} - \log(1 + \frac{d_{\min}}{m\, d_{k+1}}) \right)}_{\text{positive}} \underbrace{\left( -\frac{1}{2\epsilon}(m+1) \right)}_{\text{negative}} \tag{141}$$

By showing that $\frac{d}{d\epsilon} S_{\boldsymbol{B}_{\underline{t}-1}} < 0$ we just proved $S_{\boldsymbol{B}_{\underline{t}-1}}$ is decreasing in $\epsilon$.

We now focus on the limit case $\epsilon \to 0$. First note due to the identity $m = \frac{\|\boldsymbol{y}_0\|}{\sqrt{K}\,\epsilon} - 1$ we have the following identity,

$$\lim_{\epsilon \to 0} \min_{k \in \{1,2,\ldots,K-1\}} \left( \frac{\frac{\|\boldsymbol{y}_0\|}{\sqrt{K}\,\epsilon} - 1 + \frac{d_{\min}}{d_k}}{\frac{\|\boldsymbol{y}_0\|}{\sqrt{K}\,\epsilon} - 1 + \frac{d_{\min}}{d_{k+1}}} \right)^{\frac{\|\boldsymbol{y}_0\|}{\kappa\,\sqrt{K}\,\epsilon} - \frac{1}{\kappa}} \tag{142}$$

$$= \quad \lim_{m \to \infty} \min_{k \in \{1,2,\ldots,K-1\}} \left( \frac{m + \frac{d_{\min}}{d_k}}{m + \frac{d_{\min}}{d_{k+1}}} \right)^{\frac{1}{\kappa} m}. \tag{143}$$

Further, since pointwise minimum of continuous functions is also a continuous function, we can move the limit inside the minimum,

$$\lim_{m \to \infty} \min_{k \in \{1,2,\ldots,K-1\}} \left( \frac{m + \frac{d_{\min}}{d_k}}{m + \frac{d_{\min}}{d_{k+1}}} \right)^{\frac{1}{\kappa} m} \tag{144}$$

$$= \quad \min_{k \in \{1,2,\ldots,K-1\}} \lim_{m \to \infty} \left( \frac{m + \frac{d_{\min}}{d_k}}{m + \frac{d_{\min}}{d_{k+1}}} \right)^{\frac{1}{\kappa} m} \tag{145}$$

$$= \quad \min_{k \in \{1,2,\ldots,K-1\}} e^{\frac{\frac{d_{\min}}{d_k} - \frac{d_{\min}}{d_{k+1}}}{\kappa}} \tag{146}$$

$$= \quad \min_{k \in \{1,2,\ldots,K-1\}} e^{\frac{d_{\min}}{\kappa} (\frac{1}{d_k} - \frac{1}{d_{k+1}})} \tag{147}$$

$$= \quad e^{\frac{d_{\min}}{\kappa} \min_{k \in \{1,2,\ldots,K-1\}} (\frac{1}{d_k} - \frac{1}{d_{k+1}})}, \tag{148}$$

where in (146) we used the identity $\lim_{x \to \infty} f(x)^{g(x)} = e^{\lim_{x \to \infty} \left(f(x)-1\right)\left(g(x)\right)}$ and in (148) we used the fact that $e^{\frac{d_{\min}}{\kappa} x}$ is monotonically increasing in $x$ (because $\frac{d_{\min}}{\kappa} > 0$).

$\square$

# D More on Experiments

## D.1 Setup Details

We used Adam optimizer with learning rates of 0.001 and 0.0001 for CIFAR-10 and CIFAR-100, respectively. They are trained up to 64000 steps with batch size equal to 16 and 64 for CIFAR-10 and CIFAR-100, respectively. In all the experiments, we slightly regularize the training by weight decay regularization added to the fitting loss with its coefficient set to 0.0001 and 0.00005 for CIFAR-10 and CIFAR-100, respectively. Training and test is performed on the standard (50000 train-10000 test) split of the CIFAR dataset. Most of the experiments are conducted using Resnet-50 [12] and CIFAR-10 and CIFAR-100 datasets [18]. However, we briefly validate our results on VGG-16 [30] too.

## D.2 $\ell_2$ Loss on Neural Network Predictions

Figure 4 shows the full results on CIFAR-10 and Resnet-50. The train and test accuracies have already been discussed in the main paper and are copied here to facilitate comparison. However, in this subsection, we demonstrated the loss of the trained model at all steps with respect to the original ground truth data too. This may help establish an intuition on how self-distillation is regularizing the training on the original data. Looking at the train loss we can see it first drops as the regularization is amplified and then increases while the model under-fits. This, again, suggests that the mechanism that self-distillation employs for regularization is different from early stopping. For CIFAR-100 the results in Figure 5 show a similar trend.

Figure 4: Self distillation results with $\ell_2$ loss of neural network predictions for Resnet-50 and CIFAR-10

Figure 5: Self distillation results with $\ell_2$ loss of neural network predictions for Resnet-50 and CIFAR-100

## D.3 Self-distillation on Hard Labels

One might wonder how self-distillation would perform if we replace the neural network (soft) predictions with hard labels. In other words, the teacher's predictions are turned into one-hot-vector via `argmax` and they are treated like a dataset with augmented labels. Of-course, since the model is already over-parameterized and trained close to interpolation regime only a small fraction of labels will change. Figures 6 and 7 show the results of self distillation using cross entropy loss on labels predicted by the teacher model. Surprisingly, self-distillation improves the performance here too. This observation may be related to learning under noisy dataset and calls for more future work on this interesting case.

Figure 6: Self distillation results with cross entropy loss on hard labels for Resnet-50 and CIFAR-10

Figure 7: Self distillation results with cross entropy loss on hard labels for Resnet-50 and CIFAR-100

## D.4 Self-Distillation versus Early Stopping.

By looking at the fall of the training accuracy over self-distillation round, one may wonder if early stopping (in the sense of choosing a larger error tolerance $\epsilon$ for training) would lead to similar test performance. However, in Section 3.4 we discussed that self-distillation and early stopping have different regularization effects. Here we try to verify that. Specifically, we record the training loss value at the end of each self-distillation round. We then train a batch of models from scratch until each batch converges to one of the recorded loss values. If the regularization induced by early stopping was the same as self-distillation, then we should have seen similar test performance between a self-distilled model that achieves a specific loss value on the original training labels, and a model that stops training as soon as it reaches the same level of error. However, Figure 8 verifies that these two have different regularization effects.

Figure 8: Self-distillation compared to early stopping for Resnet50 and CIFAR-10 using $\ell_2$ and cross entropy loss, respectively.

**Self-Distillation on Other Networks.** Figure 9 shows the performance of $\ell_2$ distillation on CIFAR-100 using VGG network. This result aims to show that the theory and empirical findings are not dependent to a specific structure and apply to architectures beyond Resnet.

Figure 9: Self-distillation with $\ell_2$ loss using VGG16 Network on CIFAR-100.

## E   Mathematica Code To Reproduce Illustrative Example

```
x = (Table[i, {i, -5, 5}]/5 + 1)/2;
y = Sin[x*2*Pi] +
  RandomVariate[NormalDistribution[0, 0.5], Length[x]]
ListPlot[y]

(* UNCOMMENT IF YOU WISH TO USE EXACT SAME RANDOM SAMPLES IN THE PAPER *)
(* y = {0.38476636465198066',
  1.2333967683416893', 1.33232242218057',
  0.6920159488889518', -0.29756145531871736', -0.24189291901377769', \
-0.7964485769175675', -0.9616480167034174', -0.49672509509916934', \
-0.3469066003991437', 0.5589512650600734'}; *)

(******** PLOT GREEN'S FUNCTION g0(X,T) FOR OPERATOR d^4/dx^4 ********)

g0 = 1/6*Max[{(T - X)^3, 0}] - 1/6*T*(1 - X)*(T^2 - 2*X + X^2);
ContourPlot[g0, {X, 0, 1}, {T, 0, 1}]
Plot3D[g0, {X, 0, 1}, {T, 0, 1}]

(***** COMPUTE g AND G *****)

G = Table[
    g0 /. X -> ((i/5 + 1)/2) /. T -> ((j/5 + 1)/2), {i, -5, 5}, {j, -5,
      5}];
g = Transpose[{Table[g0 /. T -> ((j/5 + 1)/2), {j, -5, 5}]}];

(***** PLOT GROUND-TRUTH FUNCTION (ORANGE) AND OVERFIT FUNCTION \
(BLUE) *****)
FNoReg = (Transpose[g].Inverse[
      G + 0.0000000001*IdentityMatrix[Length[x]]].Transpose[{y}])[[1,
   1]];
pts = Table[{x[[i]], y[[i]]}, {i, 1, Length[x]}];
Show[{ListPlot[pts], Plot[{FNoReg, Sin[X*2*Pi]}, {X, 0, 1}]}]

(***** PARAMETERS *****)
MaxIter = 10;
eps = 0.045;

(***** SUBROUTINES *****)
Loss[G_, yin_, c_] := Module[
   {t = (G.Inverse[c*IdentityMatrix[Length[yin]] + G] -
        IdentityMatrix[Length[x]]).yin},
   Total[Flatten[t]^2]/Length[yin]
   ];

FindRootsC[f_, c_] := Module[
   {Sol = Quiet[Solve[f == 0, c]], Sel},
   Sel = Select[
     c /. Sol, (Abs[Im[#]] < 0.00000001) && # > 0.00000001 &]
```

```
    ];

(***** MAIN *****)

(* Initialization *)
y0 = Transpose[{y}];
ycur = y0;
B = IdentityMatrix[Length[x]];
FunctionSequence = {};
ASequence = {};
BSequence = {};

(* Self—Distllation Loop *)
For[i = 1;, i < MaxIter, i++,
 Print["Iteration ", i];
 Print["Norm[y]=", Norm[ycur]];
 L = Loss[G, ycur, c];
 RootsC = FindRootsC[L — eps, c];
 Switch [Length[RootsC], 0, (Print["No Root"]; Break[];), 1,
  Print["Found Unique Root c=", RootsC[[1]] ];];
 (* Now that root is unique *)
 RootC = RootsC[[1]];
 Print["Achieved Loss Value ", Loss[G, ycur, RootC]];
 U = G.Inverse[G + RootC*IdentityMatrix[Length[ycur]]];
 A = DiagonalMatrix[Eigenvalues[U]];
 f = (Transpose[g].Inverse[
      G + RootC*IdentityMatrix[Length[ycur]].ycur)[[1, 1]];
 B = B.A;
 ycur = U.ycur;

 FunctionSequence = Append[FunctionSequence, f];
 ASequence = Append[ASequence, Diagonal[A]];
 BSequence = Append[BSequence, Diagonal[B]];
 ]

If[i == MaxIter, Print["Max Iterations Reached!"]]

Plot[FunctionSequence, {X, 0, 1}]
BarChart[ASequence, ChartStyle —> "DarkRainbow", AspectRatio —> 0.2,
 ImageSize —> Full]
BarChart[BSequence, ChartStyle —> "DarkRainbow", AspectRatio —> 0.2,
 ImageSize —> Full]
```

# F    Python Implementation

Implementing self-distillation is quite straight forward provided with merely a customized loss that replaces the ground-truth labels with teacher predictions. Here, we provide a Tensorflow implementation of the self-distillation loss function:

```python
def self_distillation_loss(labels, logits, model, reg_coef,
                           teacher=None, data=None):
  if teacher is None:
    main_loss = tf.reduce_mean(tf.squared_difference(
        labels, tf.nn.softmax(logits)))
  else:
    main_loss = tf.reduce_mean(tf.squared_difference(
        tf.nn.softmax(teacher(data)), tf.nn.softmax(logits)))
  reg_loss = reg_coef*tf.add_n([tf.nn.l2_loss(w) for w
                               in model.trainable_weights])
  total_loss = main_loss + reg_loss
  return total_loss
```

The following snippet also demonstrates how one can use the above loss function to train a neural network using self-distillation.

```python
def self_distillation_train(model, train_dataset, optimizer,
                            reg_coef, epochs, teacher=None):
  for epoch in range(epochs):
    for iter, (x_batch_train,
               y_batch_train) in enumerate(train_dataset):
      with tf.GradientTape() as tape:
        logits = model(x_batch_train, training=True)
        loss_value =
        self_distillation_loss(y_batch_train, logits, model,
                               reg_coef, teacher, x_batch_train)
      grads = tape.gradient(loss_value, model.trainable_weights)
      optimizer.apply_gradients(
          zip(grads, model.trainable_weights))
  return model

teacher = None
reg_coef=1e-4
epochs=30
for step in range(distillation_steps):
  model = get_resnet_model()
  optimizer = keras.optimizers.Adam(learning_rate=learning_rate)
  model = self_distillation_train(
      model, train_dataset, optimizer, reg_coef, epochs, teacher)
  teacher = model
```

## Footnotes

[6]This is due to the conditions $\lambda_k > 0$ (recall we assume $\boldsymbol{G}$ is full-rank) and $c_i > 0$.

[7]This follows from concavity of $\frac{x}{\frac{1}{\frac{\kappa}{x-1}}+1}$ in $x$ as long as $x-1 \geq 0$ (can be verified by observing that the second derivative of the function is negative when $x-1 \geq 0$ because $\kappa > 1$ by definition). For any function $f(x)$ that is concave on the interval $[\underline{x},\overline{x}]$, any line tangent to $f$ forms an *upper* bound on $f(x)$ over $[\underline{x},\overline{x}]$. In particular, we use the tangent at the end point $\overline{x}$ to construct our bound. In our setting, this point which happens to be $r_0$. The latter is because $r_t$ is a decreasing sequence (see beginning of Section 3.2) and thus its largest values is at $t=0$.

[8]Similar to the earlier footnote, this follows from convexity of $\frac{x}{\frac{\kappa}{x-1}+1}$ in $x$ as long as $x-1 \geq 0$ since $\kappa > 1$ by definition. For any function $f(x)$ that is convex on the interval $[\underline{x},\overline{x}]$, any line tangent to $f$ forms an *lower* bound on $f(x)$ over $[\underline{x},\overline{x}]$. In particular, we use the tangent at the end point $\overline{x}$ to construct our bound, which as the earlier footnote, translate into $r_0$.

[9]More compactly, the problem can be stated as $\alpha^\dagger r_{t-1}-b \leq r_t \leq \alpha r_{t-1}-b$, where $\alpha > 0$ and $\alpha^\dagger > 0$. Let's focus on $r_t \leq \alpha r_{t-1}-b$, as the other case follows by similar argument. Start from the base case $r_1 \leq \alpha r_0-b$. Since $\alpha > 0$, we can multiply both sides by that and then add $-b$ to both sides: $\alpha r_1-b \leq \alpha^2 r_0-b(\alpha+1)$. On the other hand, looking at the recurrence $r_t \leq \alpha r_{t-1}-b$ at $t=2$ yields $r_2 \leq \alpha r_1-b$. Combining the two inequalities gives $r_2 \leq \alpha^2 r_0-b(\alpha+1)$. By repeating this argument we obtain the general case $r_t \leq \alpha^t r_0-b(\sum_{j=0}^{t-1}\alpha^j)$.