[Reviews · NeurIPS 2020]

Review 1

Summary and Contributions: This paper examines self-distillation, a popular method to improve the generalization of deep networks by re-training the network on its own predictions. Despite its popularity, self-distillation is only poorly understood and this paper attempts to provide a theoretical underpinning by examining self-distillation in Hilbert Spaces with the hope that some of the findings will provide insights to the existing practices in the deep learning community.

Strengths: The problem / topic is very timely and quite necessary. As far as I know, there exists very little prior work that tries to understand the inner workings of self-distillation.

Weaknesses: There is an obvious gap between the sub-domain where the authors can make their theoretical contributions (Hilbert spaces) and where practitioners use self-distillation. It is not clear to me how relevant the findings are to the deep learning community (who arguably are the main people using self-distillation). This gap is also present between sections 1-4 and section 5. The authors bridge this gap with a reference to the popular Neural Tangent Kernel work, but it is probably fair to say that this is a weak link. The paper falls short on its conclusion. A topic like this could really shine if there were some interesting discussion that would link back the rigorous results obtained in Hilbert Space to the ultimate domain of interest (deep learning). I am afraid the complete lack thereof diminishes the relevance of the findings quite a bit.

Correctness: I didn’t go into the nitty gritty details of the derivations, but the findings are probably correct.

Clarity: It is OK. The motivation is well stated, similarly the approach and techniques are well communicated, however a discussion of the relevance of the findings is completely missing.

Relation to Prior Work: I was a little shocked that the authors attributed model distillation to Hinton et al.’15. The well known work on “Model Compression” [Bucilua et al. 2006], precedes it by a decade.

Reproducibility: Yes

Additional Feedback: Let me state clearly that I am no learning theorist. However, still, I am quite active in the deep learning community have worked in this particular area for a while - so it is fair to expect that this paper should speak to me. —- After rebuttal: Thanks for the clarifications. I updated my review. Ultimately, I am worried that this paper is showing something inherently obvious. If I were to take a training data set, and fit it with kernelized ridge regression (very much akin to eq. (11)), then the remaining residual can be attributed to the regularization. If I repeat this process on the predictions the residual from the true label will be increased and eventually I will over-regularizer my data. There could of course be an initial benefit because my first classifier was under-regularized. This is clear, although in practice it is of course unlikely that anybody would do it, as it is easier to just set the regularization parameter better in the first place. I am sure the same applies to deep networks. What I didn’t understand is if there are implications that -for deep nets- go beyond these rather obvious findings. Although the topic is very interesting and the approach promising, all in all, I don’t think the paper should be accepted in its current form, because I believe this kind of discussion is very important in this context and very much underdeveloped in this manuscript. What I don’t know is if this could easily be added or if the theoretical contribution in itself (even without link to deep learning) is interesting enough to merit a publication. I will therefore mark this as reject with high uncertainty and am looking forward to the rebuttal and discussion with the other reviewers. Given my uncertainty I would be happy to be convinced in either direction.


Review 2

Summary and Contributions: This paper analyzes the effects of self-distillation in training neural networks. They show that self-distillation progressively reduces the number of basis functions that can be used to represent the desired solution.

Strengths: The paper shows an interesting relation between self-distillation and regularization. By limiting the number of basis functions used to represent the solution, networks trained using self-distillation can avoid over-fitting for a few rounds of training. However, if self-distillation is carried out for a "large" number of rounds, the networks can under-fit; a phenomenon also verified empirically.

Weaknesses: The paper does a very good job. This reviewer is not able to suggest any major weaknesses other than minor typos.

Correctness: Yes.

Clarity: Yes.

Relation to Prior Work: Yes.

Reproducibility: Yes

Additional Feedback:


Review 3

Summary and Contributions: The paper analyses self-distillation (knowledge distillation when the teacher and student have the same model) in the context of kernel regression, showing that self-distillation can regularize (by sparsifying the basis functions) and thus explains the performance improvement. The analysis also shows that more steps of self-distillation can lead to underfitting and thus decline in performance. These theretical insights are validated empirically on a toy example and on CIFAR10.

Strengths: 1. Novel theoretical analysis of self-distillation that can explain why it improves performance. This has been a curious phenomenon obverved and this analysis (in the context of kernel regression) yields insights on how self-distillation can sparsify the basis functions and thus regularizes the model. The paper deals with optimally tuning the regularization strength lambda in each step by formulating in terms of a fixed desired loss tolerance epsilon (equation 1). This formulation makes it easier to analyze the solution of the optimization problem in closed form. 2. Experiments validate the theory and yields interesting observations. The synthetic experiment (section 4) illustrates the theory in section 3. The experiment on CIFAR10 (section 5) also supports the theory: as one performs more steps of self-distillation, train accuracy goes down, while test accuracy goes up then down. The performance decline for large number of self-distillation steps is interesting.

Weaknesses: 1. Unclear what argument some sections are making. Section 3.1, 3.2 lower bound the number of rounds of self distillation but it is not clear what insight does it give. There seems to lack motivation for this. Section 3.5 similarly bounds the sparsity level S_B, but I'm not sure for what purpose. 2. The objective function does not correspond to distillation as done in practice. In particular, in all steps, one has both the loss with respect to the original labels y and the teacher labels. The paper only analyzes the objective with the teacher labels. If one uses the original labesl as well, I'm not sure the same phenomena (solution collapse to zero, performance decline) will happen.

Correctness: The theoretical claims appear to be correct. The empirical methodology seems correct as well.

Clarity: The dense and sparse terminologies in Section 3.3, 3.4 are a little misleading, since the entries aren't actually zero (just small). It might be better to define it clearly and relate to, for example, the intrinsic dimension of the matrix (trace divided by spectral norm).

Relation to Prior Work: Differences from previous work are sufficiently discussed.

Reproducibility: Yes

Additional Feedback: Typos: - Equation (15): $\equiv$ should be $\implies$. - line 207: nThe -> The - line 249: the closer $a$ becomes to 1 -> the closer $s$ becomes to 1? - Figure 3 caption, third line: "For Right" -> Four Right Section 3.7 mentioned "more details in appendix" but I can't find details about generalization bounds in appendix. ===== Update after authors' rebuttal: Thanks for clarifying why the self-distillation objective in the paper uses predictions and not original labels. This has addressed my concerns above, and I have updated my score.

[Author Response · NeurIPS 2020]

We thank all reviewers. It was very encouraging to read your opinion on the *strengths* of this work: importance of the problem, lack of prior work, novelty of the analysis, and consistency between our theory with empirical observation.

**[R2] Positive about this work. Minor typos.**    We are very happy that you liked our contributions and found our results intriguing. We will surely fix typos in the final version.

**[R3] In practice people use both predictions and original labels, but this paper only uses predictions.**    Thank for pointing out. In the literature, "self-distillation" refers to a host of related ideas. We have adopted the variant proposed by the well-cited work of [1], which only uses predictions. Incidentally, [1] compared training with pseudo labels generated merely from teacher's predictions versus its combination with the ground truth labels (respectively `BAN` and `BAN+L` in their Table 2), and observed that `BAN` outperforms `BAN+L`. What you are referring to is closer to `BAN+L` (unfortunately you did not specify a reference so that we could be more precise here). That being said, as self-distillation is a new area, there is not yet enough evidence to claim one variant is better than others. As such, we do not think the variant we studied is less qualified than others, if not more (due to `BAN+L` observation of [1]).

Regarding your interesting question of whether blending the original labels into the pseudo-labels can prevent collapse, it indeed does that, but at the cost of undoing some of the regularization benefits of self-distillation (consistent with `BAN+L` vs `BAN` observation of [1]). The emphasis on the original labels facilitates overfitting to those labels and diminishes the regularization effect. Note that the collapse is *not a practical concern*, as one should stop even earlier than that when over-regularization begins. The latter can be detected by measuring test performance on the validation set.

[1] T. Furlanello et al. "Born-Again Neural Networks", *ICML 2018*.

**[R3] What is the purpose of lower bound on the number of rounds of self distillation.**    To be clear, this is the lower bound on the number of self-distillation rounds before the solution *collapses* to zero. This quantity is critical in determining the ultimate strength of regularization that self-distillation can achieve. The reason is, as proved in Sections 3.3 and 3.5, the sparsity level enhances with each self-distillation round (as long as collapse is not reached) [line 188-194]. Once the solution collapses, nothing interesting happens from that point on [line 154-156]. Thus, the highest achievable sparsity is right before the collapse, which is determined by the lower bound you mentioned.

**[R3] What is the purpose of bounding $S_B$ in Section 3.5?**    The goal of the paper is to understand the implicit regularization of self-distillation. Our analysis reveals that this regularization leads to a sparser matrix $B$. $S_B$ is simply a way to quantify such sparsity; it takes the diagonal matrix $B$ and returns a single number. Since we showed the ratio between pairs of diagonals of $B$ change monotonically in $t$ [line 188-191], one can see that the sparsity index $S_B$ increases in $t$, hence showing self-distillation enhancing the sparsity. We will clarify this in the revision.

In addition, $S_B$ is used to analyze how $\epsilon$ affects the achievable sparsity level. Theorem 6 reveals that being near the interpolation regime can enhance the regularization effect of self-distillation (i.e. leading to sparser representation).

**[R1] In practice it is unlikely that anybody would do self-distillation, as it is easier to just set the regularization parameter better in the first place.**    There seems to be a *misunderstanding* here, which has caused the result to seem trivial. While it is true that one can "just set the regularization parameter better in the first place", our results show that it is *not possible* to achieve the regularization effect due to self-distillation in this way. Increasing the ridge regularization coefficient $c$ will scale all eigenvalues of the kernel by the same factor. Such uniform scaling does not change the sparsity pattern of the eigenvalues. In contrast, self-distillation exponentiates the eigenvalues by the number of steps $t$, which results in a non-uniform scaling. This allows to shrink some directions more than others. We add that increasing $c$ without any self-distillation is similar to regularization by early stopping. We have discussed this in Section 3.4 and emphasized that these two regularization schemes behave very differently.

**[R1] Is theoretical contribution of RKHS analysis alone interesting?**    Our results give insight into the behavior of self-distillation in the RKHS setting, which on its own is an interesting family of learning problems. The dynamics of self-distillation in RKHS setting follows a nonlinear recurrence for which there is no closed form solution, and this *significantly* complicates the analysis. We have been able to fully characterize the evolution of self-distillation in the RKHS setting and prove how the spectrum of the kernel evolves in an intriguing way that leads to sparsity. The fact that self-distillation promotes sparsity has *never* been noticed before, let alone proving it. We will clarify this in the revision.

**[R1] There is a gap between theory (RKHS/NTK) and practical neural networks.**    We agree that a mathematically rigorous characterization of self-distillation for deep neural networks - comparable to our results in the RKHS setting - would be an exciting contribution. However, please note that as neural networks are highly nonlinear, almost *all* theoretical developments *require* some simplifying assumptions to keep the analysis tractable.

[Meta-Review · NeurIPS 2020]

The paper was reviewed by experts on the topic and discussed after authors rebuttal. Results were found to be interesting and valuable. The reviewers comments should be taken into account while preparing the final version of the paper.